# A Vibration Fault Identification Framework for Shafting Systems of Hydropower Units: Nonlinear Modeling, Signal Processing, and Holographic Identification

**DOI:** 10.3390/s22114266

**Published:** 2022-06-03

**Authors:** Yousong Shi, Jianzhong Zhou, Jie Huang, Yanhe Xu, Baonan Liu

**Affiliations:** School of Civil and Hydraulic Engineering, Hua Zhong University of Science and Technology, Wuhan 430074, China; d201880942@hust.edu.cn (Y.S.); hjie@hust.edu.cn (J.H.); yh_xu@hust.edu.cn (Y.X.); d201981043@hust.edu.cn (B.L.)

**Keywords:** hydropower units, shafting system, nonlinear dynamic, signal multistage denoising, holospectrum, vibration fault identification

## Abstract

The shafting systems of hydropower units work as the core component for the conversion of water energy to electric energy and have been running for a long time in the hostile hydraulic–mechanical–electrical-coupled environment—their vibration faults are frequent. How to quickly and accurately identify vibration faults to improve the reliability of the unit is a key issue. This study proposes a novel shafting vibration fault identification framework, which is divided into three coordinated stages: nonlinear modeling, signal denoising, and holographic identification. A nonlinear dynamical model of bending–torsion coupling vibration induced by multiple excitation vibration sources of the shafting system is established in the first stage. The multi-stage signal denoising method combines Savitzky–Golay (SG) smoothing filtering, singular value decomposition (SVD), and variational mode decomposition (VMD). SG-SVD-VMD is used for the guide bearing the vibration signals in the second stage. Further, the holospectrum theory is innovatively introduced to obtain the holospectra of the simulated and measured signals, and the shafting vibration faults of the real unit are identified by comparing the holospectrum of the measured signal with the simulated signal. These results show that the shafting nonlinear model can effectively reflect the vibration characteristics of the coupled vibration source and reveal the influence and fault characteristics of each external excitation on the shafting vibration. The shafting vibration faults of operating units can be identified by analyzing the holospectra of the shafting simulation signals and measuring the noise reduction signals. Thus, this framework can guide the safe and stable operation of hydropower units.

## 1. Introduction

With the continuous penetration of conventional hydropower and pumped storage into the power system, hydropower, as a clean and renewable energy source, has an increasing proportion of its capacity in the power system [1], and it has become the backbone of the global dual-carbon goal [2]. Besides, the above two water energy utilization methods involve the shafting system, which is the key energy conversion component. However, the shafting of hydropower units has operated in complex and harsh hydraulic–mechanical-electrical coupling environments [3] for a long time, which greatly reduces the reliability [4] of the hydropower system and restricts the safety and efficiency of our country′s new power system. Additionally, a large number of rotating machinery failures shows obvious vibration characteristics [5]. How to quickly and accurately realize the fault diagnosis and maintenance of the shafting system has become a scientific and engineering problem that needs to be solved urgently.

In recent years, several international scholars have performed research on shafting vibration fault identification [6], which can be roughly divided into two categories. One is to establish the shafting fault mathematic model, simulate and analyze the dynamic behavior, and apply it to the engineering practice. The other is to extract features from multi-source monitoring data of shafting vibration and identify faults via machine learning [7] or deep learning [8] methods. The commonly used analyses methods for the first type include the vibration time-domain diagram, phase space trajectory, frequency spectrum diagram, Poincaré cross-section, and bifurcation diagram [4]. The common mathematical models of faults include the gyro effect [9], mass eccentricity [10], unbalanced magnetic pull (UMP) [11], rotor-stator rubbing [12], winding faults [13], shaft misalignment [14], rotor arcuate whirled (RAW) [4], hydraulic imbalance [15], seal excitation [16], guide bearing loosening [17], and turbine runner vortex eccentricity [18], etc. For the second category, firstly, the online monitoring data of shafting vibrations are denoised. The commonly used methods include ensemble empirical mode decomposition [19,20,21], SVD [22], empirical wavelet transform [23], and VMD [24,25], etc. Then, the vibration characteristic data after denoising was further extracted and input into a machine learning or deep learning algorithm for vibration fault diagnosis [26]. The common diagnostic methods are Bayesian [27], support vector machine [28], convolutional neural network [29], recurrent neural network [30], etc.

In summary, although great progress has been made in the identification of shafting vibration faults, there are still some shortcomings. There are still many shafting faults that have not been described by a mathematical model. Meanwhile, the mathematical model assumes many ideals that deviate from the real situation and fail to obtain accurate and true fault information. The second diagnosis method causes a large amount of effective information to be missing in the signal processing process, which reduces the effect of the diagnosis. Most importantly, they can only distinguish the fault types and cannot accurately identify and locate the types of faults, which greatly limits rapid maintenance and repair decisions.

To solve the above problems, a vibration fault identification framework was innovatively proposed in this study, as shown in Figure 1. The framework mainly includes three collaborative stages: nonlinear modeling, signal denoising, and vibration fault holographic identification. The innovative points mainly include: (1) the nonlinear vibration model of shafting excited by the multiple vibration sources is established, and its validity is verified by measured data. (2) The holographic spectrum theory is introduced, for the first time, in order to obtain the holospectra of the shafting vibration simulation and measure signals and identify the shafting vibration faults.

The rest of this study is organized as follows. In Section 2, a nonlinear coupling model of the shafting system with multiple vibration sources is established, and the nonlinear dynamic behavior of shafting vibration excited by multiple vibration sources is analyzed. In Section 3, we use SG-SVD-VMD to denoise the measured vibration signals of shafting in the upper, lower, and hydro-turbine bearings. In Section 4, we present the holospectrum theory method, obtain the shafting vibration two-dimensional (2D) and three-dimensional (3D) holospectra of the simulated and measured signals, and identify the shafting vibration faults. The discussion and conclusion are presented in Section 5 and Section 6, respectively.

## 2. Nonlinear Mathematical Modeling

### 2.1. Multi-Vibration Sources of the Shafting

#### 2.1.1. Rotor Stator Rubbing

The radial and tangential rubbing force of the system can be described as follows [12]:(1)$$\left\{\begin{array}{l} F_{rubx}\\ F_{ruby} \end{array}\right\}=-H(r_{1}-\delta_{0})\frac{(r_{1}-\delta_{0})k_{r}}{r_{1}}\left(\begin{array}{cc} 1 & -f\\ f & 1 \end{array}\right)\left\{\begin{array}{l} x_{1}\\ y_{1} \end{array}\right\}$$

In Equation (1), f is the friction coefficient of the hydropower unit, and $k_{r}$ is the rigidity of the stator, $\delta_{0}$ is the gap between the generator rotor and stator, $\delta_{r}=r_{1}-\delta_{0}$, where $r_{1}=\sqrt{x_{1}^{2}+y_{1}^{2}}$ is the radial displacement of a generator. H is the step function, which is shown as follows:(2)$$H\left(x\right)=\left\{\begin{array}{c} 1x\geq 0\\ 0x<0 \end{array}\right.$$

In Equation (2), when $r_{1}<\delta_{0}$, no rubbing phenomenon occurs, and the rubbing force is 0; when $r_{1}\geq \delta_{0}$, the rubbing phenomenon occurs, and the rubbing force is generated.

#### 2.1.2. UMP

Hydro-generator pole logarithms are generally greater than three teams; thus, the hydro-generator is also affected by the UMP, which is calculated as follows [11]:(3)$$F_{ump}=\beta \pi DL{\left(\frac{B}{0.5}\right)}^{2}\frac{r_{1}}{\delta_{0}}$$
in Equation (3), L and D are the generator rotor’s height and diameter, B is the magnetic density, and β is a general coefficient (β=0.2~0.5).

#### 2.1.3. Fluid Seal Excitation

The water flowing to the flange seals forms circumference flow, causing the sealing forces, which can be written as [16]:(4)$$\left\{\begin{array}{l} F_{xlf}=-\left(K-m_{f}\tau_{f}^{2}\omega^{2}\right)x_{2}-\tau_{f}\omega D_{s}y_{2}-D_{s}{\dot{x}}_{2}-2\tau_{f}m_{f}\omega {\dot{y}}_{2}-m_{f}{\ddot{x}}_{2}\\ F_{ylf}=-\left(K-m_{f}\tau_{f}^{2}\omega^{2}\right)y_{2}+\tau_{f}\omega D_{s}x_{2}-D_{s}{\dot{y}}_{2}+2\tau_{f}m_{f}\omega {\dot{x}}_{2}-m_{f}{\ddot{y}}_{2} \end{array}\right.$$
where K, $D_{s}$, $m_{f}$, and $\tau_{f}$ are the equivalent stiffness of the sealing forces, equivalent damping coefficient, equivalent mass, and nonlinear function of displacement perturbation, respectively. ω is the rotation speed.

Their corresponding expressions are as follows:(5)$$\left\{\begin{array}{l} K=K_{0}{\left(1-e_{t}^{2}\right)}^{-j}\\ D_{s}=D_{0}{\left(1-e_{t}^{2}\right)}^{-j}\\ \tau_{f}=\tau_{0}{\left(1-e_{t}\right)}^{b_{0}} \end{array}\right.$$
in Equation (5), we have $j=\frac{1}{2}∼3$, $0<b_{0}<1$; $e_{t}$ is the relative eccentricity of the turbine runner ($e_{t}=\sqrt{x_{2}^{2}+y_{2}^{2}}/\delta_{2}$, where $\delta_{2}$ is sealing clearance). j, $b_{0}$, and $\tau_{0}$ were used to describe the specific sealing; generally, we have $\tau_{0}<0.5$. The other parameters are defined in [16]. The sealing force was calculated as follows:(6)$$K_{0}=\mu_{3}\mu_{0},\;D_{0}=\mu_{1}\mu_{3}T,\;m_{f}=\mu_{2}\mu_{3}T^{2}$$

#### 2.1.4. Hydraulic Imbalance

$F_{hubx}$ and $F_{huby}$ represent the flange seals that form the circumference flow, causing sealing forces in *X* and *Y* directions, respectively, which can be written as [15]:(7)$$\left\{\begin{array}{l} F_{hubx}=-\zeta_{0}Q\frac{\rho }{A}\left(v_{2}\cos\beta_{2}-v_{1}\cos\beta_{1}\right)e_{2}^{2}\cos\phi_{2}\\ F_{huby}=-\zeta_{0}Q\frac{\rho }{A}\left(v_{2}\cos\beta_{2}-v_{1}\cos\beta_{1}\right)e_{2}^{2}\sin\phi_{2} \end{array}\right.$$

In Equation (7), $\zeta_{0}$ is the disturbance coefficient ($\zeta_{0}=0.1364$), A is the section surface ($A=\pi {\left(\frac{R_{1}+R_{2}}{2}\right)}^{2}$), $R_{1}$ is the radius at the blade′s water inlet edge, and $R_{2}$ is the radius of the lower ring at the outlet. $v_{1}$ is the flows velocity at the blade′s water inlet edge ($v_{1}=\frac{Q}{\pi R_{1}^{2}}$), $v_{2}$ is the flows velocity at the lower ring water outlet edge ($v_{2}=\frac{Q}{\pi R_{2}^{2}}$), $e_{2}$ is hydro-turbine dynamic eccentricity distance, ρ is the water density, Q is the flows, $\beta_{1}$ is flow inlet angle, and $\beta_{2}$ is flow outlet angle.

#### 2.1.5. Rotor Arcuate Whirled

The operation is accompanied by arcuate whirl centrifugal force of the generator and hydro-turbine, which were calculated as follows [4]:(8)$$\left\{\begin{array}{l} F_{Gx}=\frac{\left(J_{1}+2m_{1}e_{1}^{2}\right)\left(r_{3}+r_{4}\right)\omega^{2}}{2g}\cos\phi_{1}\\ F_{Gy}=\frac{\left(J_{1}+2m_{1}e_{1}^{2}\right)\left(r_{3}+r_{4}\right)\omega^{2}}{2g}\sin\phi_{1} \end{array}\right.$$
(9)$$\left\{\begin{array}{l} F_{Tx}=\frac{\left(J_{2}+2m_{2}e_{2}^{2}\right)\omega^{2}r_{5}}{g}\cos\phi_{2}\\ F_{Ty}=\frac{\left(J_{2}+2m_{2}e_{2}^{2}\right)\omega^{2}r_{5}}{g}\sin\phi_{2} \end{array}\right.$$

In Equations (8) and (9), $F_{Gx},F_{Gy}$ and $F_{Tx},F_{Ty}$ are the arcuate whirl centrifugal force of the generator and hydro-turbine, respectively. g is the gravitational acceleration, and ω is angular spin rate.

#### 2.1.6. Oil Film Force

The pressure distributions pertaining to the oil film forces acting on the diameters of the axes are obtained by solving the Reynolds equation. By integrating the pressure distributions of the coordinate elements, the oil film forces in *X* and *Y* directions are expressed as [31]:(10)$$\left\{\begin{array}{l} F_{oilx}=k_{xx}x+k_{xy}y+d_{xx}\dot{x}+d_{xy}\dot{y}\\ F_{oily}=k_{yx}x+k_{yy}y+d_{yx}\dot{x}+d_{yy}\dot{y} \end{array}\right.$$
in Equation (10), where the $k_{yx}$, $k_{yy}$, $k_{xx}$, and $k_{xy}$ are the oil film stiffness, and $d_{xx}$, $d_{xy}$, $d_{yx}$, and $d_{yy}$ are the oil film damping. They are written as Appendix A, and their parameters are defined in [31].

#### 2.1.7. Turbine Runner Vortex Eccentricity

We assume that the outlet ring of the turbine runner is equal to the inlet ring of the draft tube, and its expression is
(11)$$\Gamma =2\pi r_{2}V_{u2}=2\pi r_{a}V_{ua}$$

In Equation (11), $r_{2}$ is the radius of the outlet ring of the turbine runner, $V_{u2}$ is the circumferential component of the absolute velocity at the turbine runner outlet, $r_{a}$ is the radius at the midpoint of the blade′s water outlet edge, and $V_{ua}$ is the circumferential velocity at the midpoint of the blade′s water outlet edge.

The velocity triangle of the turbine runner of a hydraulic turbine shows that the velocity at the turbine runner′s outlet is
(12)$$V_{u2}=V_{2}-V_{m2}ctg\beta_{2}$$

In Equation (12), $V_{2}$ is the absolute velocity of water flow at the turbine runner outlet ($V_{2}=2\pi r_{2}n/60$, n is the speed), and $V_{m2}$ is the axial component of the turbine runner outlet ($V_{m2}=Q/F$, Q is the follows, F is turbine runner outlet water cross-section area $F=\pi r_{a}b_{d}$, and $b_{d}$ is guide vane height). So, we have:(13)$$V_{u2}=\frac{2\pi r_{2}}{60}n-\frac{Q}{F}ctg\beta_{2}$$

The rotation frequency of the vortex belt at the inlet of the draft tube can be known by definition:(14)$$f_{v}=\frac{V_{ua}}{2\pi r_{a}}=\frac{V_{u2}r_{2}^{2}}{2\pi r_{a}^{2}}={\left(\frac{r_{2}}{r_{a}}\right)}^{2}\frac{n}{60}-\frac{r_{2}}{2\pi r_{a}^{2}}\frac{Q}{F}ctg\beta_{2}$$

The hydraulic vibration source is mainly caused by the water pressure pulsation in the volute, draft tube, and action on the hydro-turbine runner [18], which is calculated as follows:(15)$$P=2\pi \rho r_{a}^{2}V_{ua}^{2}\frac{e_{v}}{r_{a}^{2}-e_{v}^{2}}$$

In Equation (15), $e_{v}$ is the vortex eccentricity distance ($e_{v}$ = 0.075), and ρ is the water density.

Assume that its pressure pulsation conforms to the following change law, we have:(16)$$\left\{\begin{array}{l} P_{x}=2\pi \rho r_{a}^{2}V_{ua}^{2}\frac{e_{v}}{r_{a}^{2}-e_{v}^{2}}\sin\left(2\pi f_{v}t\right)\cos\phi_{2}\\ P_{y}=2\pi \rho r_{a}^{2}V_{ua}^{2}\frac{e_{v}}{r_{a}^{2}-e_{v}^{2}}\sin\left(2\pi f_{v}t\right)\sin\phi_{2} \end{array}\right.$$

### 2.2. Shafting System Vibration Modeling

The shafting system structure is shown in Figure 2. As can be seen in Figure 2a, $B_{i}(i=1,2,3)$ represents the upper, lower, and hydro-turbine guide bearings. $O_{i}\left(i=1,2,3,4,5\right)$ and $r_{i}\left(i=1,2,3,4,5\right)$ represent the geometric centers and radial radii of the generator, hydro-turbine runner, and upper, lower, and hydro-turbine guide bearings, respectively. *h* is the length of the hydro-turbine shaft. We have $r_{i}=\sqrt{\left(x_{i}^{2}+y_{i}^{2}\right)}$, and we denote $\left|O_{1}O_{3}\right|=a$, $\left|O_{1}O_{4}\right|=b$, $\left|O_{4}O_{5}\right|=c$, $\left|O_{2}O_{5}\right|=d$. From the geometrical relation in Figure 2a, the following equations can be obtained [4,32,33]:(17)$$r_{3}=\frac{\left[\left(a+b\right)\left(b+c+d\right)-a\left(c+d\right)\right]r_{1}+abr_{2}}{b(b+c+d)}$$
(18)$$r_{4}=\frac{\left(c+d\right)r_{1}+br_{2}}{b+c+d}$$
(19)$$r_{5}=\frac{dr_{1}+\left(b+c\right)r_{2}}{b+c+d}$$

Figure 2b shows that the misalignment coordinates are as follows:(20)$$\left\{\begin{array}{l} x_{2}=x_{1}+\left|O_{1}O_{2}\right|\cos\theta =x_{1}+h\sin\varphi \cos\theta \\ y_{2}=y_{1}+\left|O_{1}O_{2}\right|\sin\theta =y_{1}+h\sin\varphi \sin\theta \end{array}\right.$$

As can be seen in Figure 2c, the generator rotor and hydro-turbine runner coordinate relationship are written as follows:(21)$$\left\{\begin{array}{l} x_{c1}=x_{1}+e_{1}\cos\phi_{1}\\ y_{c1}=y_{1}+e_{1}\sin\phi_{1} \end{array}\right.$$
(22)$$\left\{\begin{array}{l} x_{c2}=x_{2}+e_{2}\cos\varphi \cos\phi_{2}\\ y_{c2}=y_{2}+e_{2}\cos\varphi \sin\phi_{2} \end{array}\right.$$

Assuming that the hydro-turbine and generator spindles are connected rigidly, the kinetic energy of the shafting system can be written as:(23)$$\begin{array}{c} T=\frac{m_{1}\left({\dot{x}}_{1}^{2}+{\dot{y}}_{1}^{2}+e_{1}^{2}{\dot{\phi }}_{1}^{2}+2e_{1}{\dot{\phi }}_{1}{\dot{y}}_{1}\cos\phi_{1}-2e_{1}{\dot{\phi }}_{1}{\dot{x}}_{1}\sin\phi_{1}\right)}{2}+\frac{\left(J_{2}+m_{2}{\left(e_{2}\cos\varphi +h\sin\varphi \right)}^{2}\right){\dot{\phi }}_{2}^{2}}{2}+\\ \frac{\left(J_{1}+m_{1}e_{1}^{2}\right){\dot{\phi }}_{1}^{2}}{2}+\frac{m_{2}\left({\dot{x}}_{2}^{2}+{\dot{y}}_{2}^{2}+{\left(e_{2}\cos\varphi \right)}^{2}{\dot{\phi }}_{2}^{2}+2e_{2}\cos\varphi {\dot{\phi }}_{2}{\dot{y}}_{2}\cos\phi_{2}-2e_{2}\cos\varphi {\dot{\phi }}_{2}{\dot{x}}_{2}\sin\phi_{2}\right)}{2} \end{array}$$

In Equation (23), $\left(x_{i},\left.y_{i}\right)\right.$,$J_{i}$,$m_{i}$,$e_{i}$,$\phi_{i}$, and ${\dot{\phi }}_{i}=\omega_{i}$$\left(i=1,2\right)$ are the centroid coordinates, rotational inertia, equivalent concentrated masses, eccentricity distance, rotation angle, and rotational angular velocity of the generator rotor and hydro-turbine runner, respectively.

Considering the flexural–torsional-coupled vibrations, the potential energy of the shafting can be expressed as follows:(24)$$U=\frac{k_{1}\left(x_{1}^{2}+y_{1}^{2}\right)}{2}+\frac{k_{2}\left(x_{2}^{2}+y_{2}^{2}\right)}{2}+\frac{k_{y}{\left(\phi_{1}-\phi_{2}\right)}^{2}}{2}$$

In Equation (24), we have:(25)$$\left\{\begin{array}{l} k_{1}=2K_{11}+K_{12}\;r_{2}/r_{1}\\ k_{2}=2K_{22}+K_{12}\;r_{1}/r_{2} \end{array}\right.$$
where $K_{11}$, $K_{22}$, and $K_{12}$ is the equivalent stiffness of the shafting system, and $k_{y}$ is the elastic coefficient of the spindle.
(26)$$\left\{\begin{array}{l} K_{11}=\frac{A_{1}^{2}}{B^{2}}k_{3}+\frac{{\left(c+d\right)}^{2}}{{\left(b+c+d\right)}^{2}}\begin{array}{cc} k_{4}+\frac{d^{2}}{{\left(b+c+d\right)}^{2}}k_{5} & \left(a\right) \end{array}\\ K_{22}=\frac{A_{2}^{2}}{B^{2}}k_{3}+\frac{b^{2}}{{\left(b+c+d\right)}^{2}}\begin{array}{cc} k_{4}+\frac{{\left(b+c\right)}^{2}}{{\left(b+c+d\right)}^{2}}k_{5} & \left(b\right) \end{array}\\ K_{12}=\frac{-A_{1}A_{2}}{B^{2}}k_{3}+\frac{b\left(c+d\right)}{{\left(b+c+d\right)}^{2}}\begin{array}{cc} k_{4}+\frac{d\left(b+c\right)}{{\left(b+c+d\right)}^{2}}k_{5} & \left(c\right) \end{array} \end{array}\right.$$
in Equation (26), $A_{1}=\left(a+b\right)\left(b+c+d\right)$, $A_{2}=ab$, B=b(b+c+d), $k_{3}$, $k_{4}$, and $k_{5}$ are the supporting stiffness of the upper, lower, and hydro-turbine guide bearings, respectively [4].
(27)$$k_{y}=\frac{E\pi \left(d_{H}^{2}-d_{B}^{2}\right)}{32l}$$

In Equation (27), E is the elastic modulus of the spindle. $d_{H}$ and $d_{B}$ are the internal and external diameters of the spindle. l is the total length of the generator’s lower and hydro-turbine spindles.

Assuming that the shafting damping is simplified as linear damping applied to the generator and hydro-turbine runner, and the external excitation force is also considered, then the generalized force can be formulated as follows [32,33]:(28)$$\left\{\begin{array}{l} Q_{xi}=-c_{i}x_{i}+F_{xi}\\ Q_{yi}=-c_{i}y_{i}+F_{yi} \end{array}\right.$$

In Equation (28), $c_{i}$ is the damping coefficient, $F_{xi}$ is the excitation force in the horizontal x direction, and $F_{yi}$ is the excitation force in the horizontal y direction.

Lagrange equation can be expressed as [32,33]:(29)$$\frac{d}{dt}\left(\frac{\partial T}{\partial {\dot{q}}_{i}}\right)-\left(\frac{\partial T}{\partial q_{i}}\right)+\left(\frac{\partial U}{\partial q_{i}}\right)=Q_{i}$$
where T is the kinetic energy expressed by various generalized coordinates and velocities of the system. U is the potential energy represented by the generalized coordinates. $Q_{i}$ is the corresponding generalized force.

It is considered that the generator is mainly stimulated by generator rotor–stator rub impact, UMP, rotor arcuate rotation, etc. Meanwhile, it can be seen in Figure 2a that the upper and lower guide bearings are close to the generator rotor, so the reaction force of the oil film force of the upper and lower guide bearings ($F_{oilxu}$,$F_{oilxl}$,$F_{oilyu}$,$F_{oilyl}$) is approximately equivalent to that on the generator rotor. The hydro-turbine runner is mainly excited by seal excitation force, hydraulic imbalance force, vortex belt eccentric force, turbine runner arcuate rotation, etc. Meanwhile, it can be seen in Figure 2a that the hydro-turbine guide bearing is close to the turbine runner, so the reaction force of the oil film force at the hydro-turbine guide bearing ($F_{oilxh}$,$F_{oilyh}$) is approximately equivalent to the turbine runner.

To sum up, when substituting Equations (23), (24) and (28) into Equation (29), the vibration motion equations of the shafting can be written as:
(30)$$\left\{\begin{array}{cc} \; & m_{1}{\ddot{x}}_{1}+c_{1}{\dot{x}}_{1}-m_{1}e_{1}{\ddot{\phi }}_{1}\sin\phi_{1}+k_{1}x_{1}-m_{1}e_{1}{\dot{\phi }}_{1}^{2}\cos\phi_{1}=F_{rubx}+F_{umpx}+F_{Gx}+F_{oilxu}+F_{oilxl}\begin{array}{cc} & \left(a\right) \end{array}\\ \; & m_{1}{\ddot{y}}_{1}+c_{1}{\dot{y}}_{1}+m_{1}e_{1}{\ddot{\phi }}_{1}\cos\phi_{1}+k_{1}y_{1}-m_{1}e_{1}{\dot{\phi }}_{1}^{2}\sin\phi_{1}=F_{ruby}+F_{umpy}+F_{Gy}+F_{oilyu}+F_{oilyl}\begin{array}{cc} & \left(b\right) \end{array}\\ \; & m_{2}{\ddot{x}}_{2}+c_{2}{\dot{x}}_{2}-m_{2}e_{2}\cos\varphi {\ddot{\phi }}_{2}\sin\phi_{2}+k_{2}x_{2}-m_{2}e_{2}\cos\varphi {\dot{\phi }}_{2}^{2}\cos\phi_{2}=F_{xlf}+F_{hubx}+F_{Tx}+F_{oilxh}+P_{x}\begin{array}{cc} & \left(c\right) \end{array}\\ \; & m_{2}{\ddot{y}}_{2}+c_{2}{\dot{y}}_{2}+m_{2}e_{2}\cos\varphi {\ddot{\phi }}_{2}\cos\phi_{2}+k_{2}y_{2}-m_{2}e_{2}\cos\varphi {\dot{\phi }}_{2}^{2}\sin\phi_{2}=F_{ylf}+F_{huby}+F_{Ty}+F_{oilyh}+P_{y}\begin{array}{cc} & \left(d\right) \end{array}\\ \; & \left(J_{1}+2m_{1}e_{1}^{2}\right){\ddot{\phi }}_{1}+m_{1}e_{1}{\ddot{y}}_{1}\cos\phi_{1}-m_{1}e_{1}{\ddot{x}}_{1}\sin\phi_{1}+k_{y}\alpha =-M_{g}\begin{array}{cc} & \left(e\right) \end{array}\\ \; & \left(J_{2}+2m_{2}{\left(e_{2}\cos\varphi +h\sin\varphi \right)}^{2}\right){\ddot{\phi }}_{2}+m_{2}e_{2}\cos\varphi {\ddot{y}}_{2}\cos\phi_{2}-m_{2}e_{2}\cos\varphi {\ddot{x}}_{2}\sin\phi_{2}-k_{y}\alpha =M_{t}\begin{array}{cc} & \left(f\right) \end{array} \end{array}\right.$$

In Equation (30), $M_{t}=m_{t}M_{gB}$, $M_{g}=m_{g}M_{gB}$ are the hydro-turbine torque and generator torque, respectively. ${\dot{\phi }}_{1}={\dot{\phi }}_{2}=\omega_{1}=\omega_{2}=\omega$ is the inherent coaxial rotation, $\phi_{1}$ is the rotational phase of the generator rotor, and $\phi_{2}$ is the rotational phase of the hydro-turbine runner. We have $\phi_{1}=\int \omega dt+\varphi_{1}$ and $\phi_{2}=\int \omega dt+\varphi_{2}$, and it can be further determined that the phase difference between the generator and hydro-turbine runner is $\alpha =\phi_{1}-\phi_{2}=\varphi_{1}-\varphi_{2}$

In Equation (30), we have $\left(J_{2}+2m_{2}(e_{2}\cos\varphi +h\sin\varphi )e_{2}^{2}\right)\times (e)-\left(J_{1}+2m_{1}e_{1}^{2}\right)\times (f)$, which can be obtained via [4]:(31)$$\begin{array}{c} \ddot{\alpha }=-c_{t}\dot{\alpha }-\frac{m_{1}e_{1}{\ddot{y}}_{1}\cos\phi_{1}-m_{1}e_{1}{\ddot{x}}_{1}\sin\phi_{1}}{\left(J_{1}+2m_{1}e_{1}^{2}\right)}+\frac{m_{2}e_{2}{\ddot{y}}_{2}\cos\phi_{2}-m_{2}e_{2}{\ddot{x}}_{2}\sin\phi_{2}}{\left(J_{2}+2m_{2}{\left(e_{2}\cos\varphi +h\sin\varphi \right)}^{2}\right)}-\frac{M_{g}}{J_{1}+2m_{1}e_{1}^{2}}-\\ \frac{Jk_{y}\alpha }{\left(J_{2}+2m_{2}{\left(e_{2}\cos\varphi +h\sin\varphi \right)}^{2}\right)\left(J_{1}+2m_{1}e_{1}^{2}\right)}-\frac{M_{t}}{\left(J_{2}+2m_{2}{\left(e_{2}\cos\varphi +h\sin\varphi \right)}^{2}\right)} \end{array}$$
where $J=J_{2}+2m_{2}{\left(e_{2}\cos\varphi +h\sin\varphi \right)}^{2}+J_{1}+2m_{1}e_{1}^{2}$, and $c_{t}$ is the torsional damping coefficient, since the hydropower unit has a coaxial rotation, ${\ddot{\phi }}_{1}={\dot{\omega }}_{1}={\ddot{\phi }}_{2}={\dot{\omega }}_{2}=\dot{\omega }$$\omega_{1}=\omega_{2}=\omega$. Therefore, the vibration equation of the shafting can be obtained as follows:
(32)$$\left\{\begin{array}{cc} \; & {\ddot{x}}_{1}=\left(F_{rubx}+F_{umpx}+F_{Gx}+F_{oilxu}+F_{oilxl}-c_{1}{\dot{x}}_{1}-\left(2K_{11}+\frac{\sqrt{x_{2}^{2}+y_{2}^{2}}}{\sqrt{x_{1}^{2}+y_{1}^{2}}}K_{12}\right)x_{1}+m_{1}e_{1}\omega^{2}\cos\phi_{1}\right)/m_{1}\\ \; & {\ddot{y}}_{1}=\left(F_{ruby}+F_{umpy}+F_{Gy}+F_{oilyu}+F_{oilyl}-c_{1}{\dot{y}}_{1}-\left(2K_{11}+\frac{\sqrt{x_{2}^{2}+y_{2}^{2}}}{\sqrt{x_{1}^{2}+y_{1}^{2}}}K_{12}\right)y_{1}+m_{1}e_{1}\omega^{2}\sin\phi_{1}\right)/m_{1}\\ \; & {\ddot{x}}_{2}=\left(F_{xlf}+F_{hubx}+F_{Tx}+F_{oilxh}+P_{x}-c_{2}{\dot{x}}_{2}-\left(2K_{22}+\frac{\sqrt{x_{1}^{2}+y_{1}^{2}}}{\sqrt{x_{2}^{2}+y_{2}^{2}}}K_{12}\right)x_{2}+m_{2}e_{2}\cos\varphi \omega^{2}\cos\phi_{2}\right)/m_{2}\\ \; & {\ddot{y}}_{2}=\left(F_{ylf}+F_{huby}+F_{Ty}+F_{oilyh}+P_{y}-c_{2}{\dot{y}}_{2}-\left(2K_{22}+\frac{\sqrt{x_{1}^{2}+y_{1}^{2}}}{\sqrt{x_{2}^{2}+y_{2}^{2}}}K_{12}\right)y_{2}+m_{2}e_{2}\cos\varphi \omega^{2}\sin\phi_{2}\right)/m_{2}\\ \; & \ddot{\alpha }=-c_{t}\dot{\alpha }-\frac{m_{1}e_{1}{\ddot{y}}_{1}\cos\phi_{1}-m_{1}e_{1}{\ddot{x}}_{1}\sin\phi_{1}}{\left(J_{1}+2m_{1}e_{1}^{2}\right)}+\frac{m_{2}e_{2}{\ddot{y}}_{2}\cos\phi_{2}-m_{2}e_{2}{\ddot{x}}_{2}\sin\phi_{2}}{\left(J_{2}+2m_{2}{\left(e_{2}\cos\varphi +h\sin\varphi \right)}^{2}\right)}-\frac{M_{g}}{J_{1}+2m_{1}e_{1}^{2}}-\\ \; & \frac{Jk_{y}\alpha }{\left(J_{2}+2m_{2}{\left(e_{2}\cos\varphi +h\sin\varphi \right)}^{2}\right)\left(J_{1}+2m_{1}e_{1}^{2}\right)}-\frac{M_{t}}{\left(J_{2}+2m_{2}{\left(e_{2}\cos\varphi +h\sin\varphi \right)}^{2}\right)} \end{array}\right.$$

According to Equation (32), we can solve the shafting dynamic vibration responses $x_{1}$, $y_{1}$, $x_{2}$, $y_{2}$, and α. Further, according to Equations (17)–(19), the vibration dynamic response at the guide bearing can be calculated as follows:(33)$$\left\{\begin{array}{l} x_{3}=\frac{\left[\left(a+b\right)\left(b+c+d\right)-a\left(c+d\right)\right]r_{1}+abr_{2}}{b(b+c+d)}\cos\phi_{1}\\ y_{3}=\frac{\left[\left(a+b\right)\left(b+c+d\right)-a\left(c+d\right)\right]r_{1}+abr_{2}}{b(b+c+d)}\sin\phi_{1} \end{array}\right.$$
(34)$$\left\{\begin{array}{l} x_{4}=\frac{\left(c+d\right)r_{1}+br_{2}}{b+c+d}\cos\phi_{1}\\ y_{4}=\frac{\left(c+d\right)r_{1}+br_{2}}{b+c+d}\sin\phi_{1} \end{array}\right.$$
(35)$$\left\{\begin{array}{l} x_{5}=\frac{dr_{1}+\left(b+c\right)r_{2}}{b+c+d}\cos\phi_{2}\\ y_{5}=\frac{dr_{1}+\left(b+c\right)r_{2}}{b+c+d}\sin\phi_{2} \end{array}\right.$$

### 2.3. Shafting System Vibration Nonlinear Dynamical Behavior

In Section 2.2, the nonlinear mathematical model excited by multi-vibration sources of the shafting system has been established, and its nonlinear vibration dynamic behavior plays a significant role in vibration fault identification. Therefore, the classical fourth-order Runge–Kutta method is adopted to solve the nonlinear equation of the shafting system in MATLAB. The actual design parameters of a hydropower station in China are shown in Table 1.

#### 2.3.1. Dynamic Behavior Analysis of the Rated Condition

The shafting vibrations dynamic characteristics of the generator and hydro-turbine runner are shown in Figure 3. As can be seen in Figure 3, the axis track of generator vibration is composed of multiple complex closed curves, Poincare section is composed of many irregular points, and vibration frequency appears as multiple frequency components. It can be qualitatively analyzed that the generator performs complex multi-frequency, quasi-periodic vibration under the rated working conditions, and its main vibration frequency is 0.1591 octaves. Similarly, we can qualitatively analyze that the hydro-turbine runner is a complex quasi-periodic vibration, and its main vibration frequency component is concentrated in 0~0.5 octaves. Meanwhile, the vibration amplitude of the hydro-turbine runner is much smaller than the generator, and the vibration frequency component of the hydro-turbine runner is more complex than the generator. Therefore, we conclude that the complex vibration law of the hydro-turbine runner is caused by the flow around the fluid. As can be seen in Figure 4a,b, the vibration characteristics of the upper and lower guide bearings are almost the same, and their trajectory is composed of multiple closed curves. The Poincare section contains multiple discrete points; the main frequency component is 0.1591 octaves and contains some frequency components with a small amplitude of 0~1 octaves. It can be seen in Figure 4c that the hydro-turbine guide bearing vibration trajectory is composed of multiple closed curves. The Poincare section contains multiple discrete points; the main frequency component is 0.1591 octaves and contains some frequency components with a small amplitude of 0~0.5 octaves. By comparing the vibration tracks of the hydro-turbine guide bearing with the upper and lower guide bearings, it can be found that the vibration characteristics of the hydro-turbine guide bearing are more complex than the upper and lower guide bearings, hydro-turbine guide bearing vibration amplitude is smaller than the upper and lower guide bearings, and vibration frequency components of the hydro-turbine guide bearing are greater.

The dynamic behavior of the torsional vibration of the shaft is shown in Figure 5. As can be seen in Figure 5, the shaft torsional amplitude is about 0.013 (rad). The Poincare cross-section is composed of many irregular points at a small value of torsional amplitude; thus, the torsional vibration can almost be ignored as stable. Meanwhile, the stable characteristics of torsional vibration can also be seen from the spectrum of the torsional vibration.

#### 2.3.2. Nonlinear Dynamic Behavior Analysis of Shafting Vibration under Variable Speed

The change of rotation speed is the most obvious in the process of the peak and frequency regulation of conventional hydropower units, especially variable speed pumped storage units. Therefore, it is significantly important to study the vibration dynamic behavior of the shafting system under variable speed conditions. The nonlinear dynamic behavior of shafting system vibrations of variable speed is shown in Figure 6. As can be seen in Figure 6a, when ω is $\left[3.98,28.9\right)$ rad/s, the generator vibrations are quasi-periodic, which shows obvious vibration and strong sensitivity, and the vibration amplitude increases gradually. When ω is $\left[0,3.98\right)$ and $\left[28.9,32\right]$ rad/s, the generator vibrations are periodic, and the sensitivity to ω in this range is almost negligible, but the vibration amplitude increases gradually. As can be seen in Figure 6b, when ω is $\left[0,28.69\right)$ rad/s, the hydro-turbine vibrations are quasi-periodic, which shows obvious vibration and strong sensitivity, and the vibration amplitude decreases gradually. When ω is $\left[28.69,32\right]$ rad/s, the hydro-turbine vibrations remain stable, the vibration amplitude remains samely, and the sensitivity to ω in this range is almost negligible. As can be seen in Figure 6c,d, when ω is $\left[0,28.69\right)$ rad/s, the upper and lower guide bearing vibrations are quasi-periodic, which show obvious vibration and strong sensitivity. When ω is $\left[28.69,32\right]$ rad/s, the generator vibrations remain stable, and the sensitivity to ω in this range is almost negligible. Additionally, when ω is $\left[0,14.87\right]$ rad/s, the vibration amplitude decreases gradually; however, when ω is $\left[14.87,32\right]$ rad/s, the vibration amplitude is creasing gradually. As can be seen in Figure 6e, when ω is $\left[0,28.9\right)$ rad/s, the hydro-turbine guide bearing vibrations are quasi-periodic, which shows obvious vibration and strong sensitivity, and the vibration amplitude decreases gradually. When ω is $\left[28.9,32\right]$ rad/s, the hydro-turbine guide bearing vibrations remain stable, vibration amplitude increases, and sensitivity to ω in this range is almost negligible. As can be seen in Figure 6f, when ω is $\left[0,28.69\right)$ rad/s, the shaft torsional vibrations are quasi-periodic, which shows obvious vibration and strong sensitivity. When ω is $\left[28.69,32\right]$ rad/s, the shaft torsional vibrations are quasi-periodic, which shows slight vibration, and the sensitivity to ω in this range is almost negligible. By comparing the variable speed of the vibration bifurcation diagrams of the upper and lower guide bearings and generator, it can be found that they have almost the same nonlinear vibration characteristics. By comparing the variable speed of the vibration bifurcation diagrams of hydro-turbine runner and hydro-turbine guide bearing, it can be found that they have almost the same nonlinear vibration characteristics.

## 3. SG-SVD-VMD Fusion Signal Denoising

In this section, we use the SG-SVD-VMD [34,35] multistage denoising method to denoise the actual measured signals of the upper, lower, and hydro-turbine bearings of a power station, same as in Section 2.3. The specific process is shown in Figure 7.

The length of the moving window is 41. Meanwhile, set the number of rows for the reconstructed Hankel matrix to 8. The signal-to-noise ratio (SNR) and root-mean-square error (RMSE) of each method for shafting guide bearing actual measured signal were shown in Table 2. As can be seen in Table 2, after SG-SVD-VMD denoising, the shafting vibration data has higher SNR and smaller RMSE. The measured data of the shafting vibration of a power station were denoised at multiple levels; the data before and after denoising were shown in Figure 8, and the spectrum diagram before and after shafting vibration denoising is shown in Figure 9. Meanwhile, as can be seen in Figure 8 and Figure 9, via the shafting vibration condition monitoring data by SG-SVD-VMD multi-stage noise reduction, some noise components have been filtered out.

## 4. Holospectrum Diagnostic Technology

The holospectrum was developed based on the Fourier spectrum. Through the Fourier transform of two mutually perpendicular time-domain signals in the corona measuring plane, the amplitude spectrum was synthesized in the frequency domain, according to a certain algorithm; then, the 2D holospectrum of the signal could be obtained [36].

### 4.1. The 2D Holospectrum

The principle of the 2D holospectrum [37] is shown in Figure 10. It not only reflects the vibration amplitude of the rotor in two directions but also the phase relationship between them.

### 4.2. The 3D Holospectrum

The 3D holospectrum [38] is composed of the elliptic spectrum of each order frequency, corresponding to each node of the rotation axis, as shown in Figure 11. The IPP of the ellipse on each node is connected by the original creation line. Similarly, the other points on the ellipse are also creatively connected, in order of sampling time. Therefore, the 3D holospectrum shows the vibration status of all nodes of the rotating shaft, and the corresponding points on the generated line represent the vibration status of each node at the corresponding sampling time. The main components of the 3D holospectrum are the frequency ellipse, initial phase points on the ellipse, and formation lines connecting the initial phase points.

### 4.3. Shafting Vibration Fault Identification

As shown in Figure 1, due to the complex installation of the generator rotor and hydro-turbine runner, the current monitoring technology is not perfect. At present, the vibration dynamic response of the shafting system is mainly characterized by monitoring the vibration of the guide bearings. Meanwhile, according to mathematical equation—Equations (33)–(35), the vibrations of the guide bearings are simulated.

To identify the vibration fault of hydropower unit shafting, we successively consider the external excitation in Section 2.1, simulate the vibration response of guide bearings, obtain their spectra, i.e., the 2D and 3D holographic spectra of each external excitation action, and analyze the fault characteristics of each external excitation. Further, the fusion SG-SVD-VMD multistage denoising method is used to denoise the actual monitoring signals of the guide bearings and their spectra, i.e., the 2D and 3D holographic spectra, which are obtained by using holographic theory. Finally, the shafting vibration fault of the actual unit is identified by comparing the spectra, i.e., the 2D and 3D holographic spectra, of the measured and simulation signals of the external excitation sources.

According to the holospectrum theory method, we obtained the spectra, i.e., the 2D and 3D holospectra, which successively increased, considering the external excitation effects, such as rotor unbalance, shaft misalignment, oil film force, RAW, rotor–stator RUB, UMP, turbine runner vortex eccentricity, hydraulic imbalance force, and seal excitation force, which are shown in Figure 12, Figure 13, Figure 14, Figure 15, Figure 16, Figure 17, Figure 18, Figure 19 and Figure 20, respectively.

As can be seen in Figure 12, the vibration frequency of guide bearings under the rotor unbalance excitation is 0.1554 octaves, and the vibration amplitude of the upper guide bearing is the largest, followed by the lower and turbine guide bearings. The 2D holospectra of the upper, lower, and hydro-turbine guide bearings mainly includes the fundamental frequency and 2 octaves. The line created by the 3D holospectrum is straight, without crossing, which indicated it is not affected by the force couple.

By comparing Figure 13 with Figure 12, it is found that the vibration frequency characteristic of guide bearing and the 2D and 3D holospectra almost do not change when the spindle misalignment is considered. It can be qualitatively analyzed that the shaft misalignment has little influence on the guide bearing’s vibrations.

By comparing Figure 14 with Figure 13, it is found that the vibration frequency characteristic of the guide bearings and 2D and 3D holospectra hardly change when the external excitation of oil film force is considered. It can be qualitatively analyzed that the oil film force has little influence on the vibrations of guide bearings and mainly plays a lubrication role.

Comparing Figure 15 with Figure 14, it is found that, when the external excitation of RAW is taken into account, the vibration frequency characteristic of guide bearings remains unchanged, but the vibration amplitude increases significantly, and the 2D and 3D holospectra hardly change. It can be qualitatively analyzed that RAW increased guide bearings vibrations amplitude.

Comparing Figure 16 with Figure 15 showed that the vibration frequency characteristics and 2D and 3D holospectra of the guide bearings almost do not change when the external excitation of the rotor–stator rub force is considered, and it can be qualitatively analyzed that no rotor–stator rub occurs.

By comparing Figure 17 with Figure 16, it is found that the vibrations frequency characteristics of the guide bearings and 2D and 3D holospectra almost do not change when the UMP external excitation is considered. It can be qualitatively analyzed that the UMP has almost no influence on the vibrations of guide bearings.

By comparing Figure 18 with Figure 17, we found that, when considering increased turbine runner vortex eccentricity, the guide bearings vibrations frequency components significantly increased, and the upper and lower guide bearing vibrations amplitude decreases, but the hydro-turbine guide bearing vibrations amplitude was increased. Additionally, the upper and lower guide bearing’s 2D holospectrum includes fundamental, 2 octaves, and triple frequency. The hydro-turbine guide bearing includes the 1~8 octave frequency components. The intersecting lines of the lower guide bearing and hydro-turbine guide bearing in the 3D holospectrum indicate that there are a couple of imbalances. It can be qualitatively analyzed that the turbine runner vortex eccentricity has a significant influence on guide bearings vibrations.

By comparing Figure 19 with Figure 18, the guide bearings vibrations frequency characteristic and 2D and 3D holospectra are almost unchanged when the external excitation of hydraulic imbalance is considered; therefore, it can be qualitatively analyzed that the hydraulic imbalance force has little influence on guide bearings vibrations.

By comparing Figure 20 with Figure 19, it is found that the vibrations frequency characteristic amplitude of guide bearings decreases slightly when the external excitation of sealing excitation is taken into account, but the 2D and 3D holospectra hardly changed. It can be qualitatively analyzed that the sealing excitation has little influence on the vibrations of guide bearings.

By comparing the spectruma, i.e., the 2D and 3D holospectra, under various excitation in Figure 12, Figure 13, Figure 14, Figure 15, Figure 16, Figure 17, Figure 18, Figure 19 and Figure 20, we find that the rotor imbalance, RAW, and turbine runner vortex belt eccentricity have the most significant impact on the shafting vibration of the hydropower unit. Rotor unbalance is the main vibration source, and its excitation vibration amplitude is the largest. The RAW is mainly reflected in the increasing of the vibration amplitude of the shafting, and the eccentricity of the turbine runner vortex is mainly reflected in the increasing of a variety of vibration frequency components and via exciting the torque imbalance.

The guide bearings 2D and 3D holospectra of the measured signals are shown in Figure 21. As can be seen in Figure 21, the upper and lower guide bearings have a fundamental frequency and small 2 octaves, in addition to 7 and 8 octaves. There are obvious 1–8 octave components in the hydro-turbine guide bearing, and the cross of the formation lines of the lower and hydro-turbine guide bearing in the 3D holospectrum indicates there are a couple imbalances. Based on a comprehensive contrast of Figure 9b, Figure 21 and Figure 12, Figure 13, Figure 14, Figure 15, Figure 16, Figure 17, Figure 18, Figure 19 and Figure 20, the upper and lower guide bearings 2D holospectrum of the measurement and simulation signal have the same fundamental frequency and smaller 2 octaves, and the stimulation signal frequency components are produced by the rotor unbalance force. The hydro-turbine guide bearing 2D holospectrum of the measurement and simulation signal has the same 1–5 octave frequency, and the stimulation signal frequency components are produced by turbine runner vortex eccentricity and the lower and hydro-turbine guide bearings 3D holospectra of the cross line. Therefore, the faults of rotor unbalance, turbine runner vortex belt eccentricity, and force couple unbalance can be qualitatively analyzed in real units.

## 5. Discussion

In combination with Figure 2a, Figure 3, Figure 4a,b, and Figure 6a,c,d, it can be seen that the upper and lower guide bearings and generator are close to each other, and they have similar nonlinear vibration dynamic behavior. As can be seen in Figure 2a, Figure 3, Figure 4c and Figure 6b,e, the hydro-turbine runner and hydro-turbine guide bearing are close to each other, and they have similar vibration dynamics behavior. Further, as can be seen in Figure 22, the vibration amplitude of the simulation and actual measurement signal of the guided bearings are close to each other, and they all show complex quasi-periodic vibrations. Additionally, the vibration trajectory of the hydro-turbine guide bearing is more complicated than the upper and lower guide bearings. It can be preliminarily inferred that the main reason for this phenomenon is that the hydro-turbine guide bearing is close to the turbine runner and affected by complex hydraulic factors. As can be seen in Figure 9b and Figure 20a, the measured and simulated signal have the same main frequency, namely 0.16 and 0.31 octaves. In summary, we can analyze that the shafting vibration nonlinear model can effectively reveal the vibration characteristics.

## 6. Conclusions

This study presents a shafting vibration fault identified framework, which includes three stages: nonlinear modeling, signal denoise, and holographic identification. A nonlinear dynamical model of bending–torsion coupling vibration induced by multiple vibration sources of the shafting system was proposed in the first stage. The multi-stage signal denoise method SG-SVD-VMD was used for guide bearings nonlinear and non-stationary vibration signals in the second stage. Further, the holospectrum theory was innovatively introduced for shafting vibration fault diagnosing. There are some major conclusions, as follows:(1)In the low-speed region, the shafting vibrations that are excited by multiple vibration sources are complex and quasi-periodic, which show obvious vibration, strong sensitivity, and accompany some frequency components.(2)The rotor unbalances, RAW, and turbine runner vortex eccentricity have the greatest influence on shafting vibrations. The influence of rotor imbalance and RAW are mainly reflected in the increasing vibration amplitude, and the influence of turbine runner vortex eccentricity is mainly reflected in the increasing vibration frequency components and by exciting the torque imbalance.(3)The rotor unbalances, turbine runner vortex eccentricity, and couple unbalance vibration faults in a real operating unit were identified by comparing the measured signal holospectrum with the simulation signal holospectrum.(4)The shafting vibration fault identification framework verified that the nonlinear mathematical model of shafting vibration is effective, and it can obtain results near that of the real dynamic behavior of shafting vibration and realize signal noise reduction. More importantly, the shafting vibration faults were quickly and effectively identified by the holospectrum, which lays a theoretical foundation for the safe and stable operation of the hydropower units.

To sum up, there are still some shortcomings in this study. In future research, we should further explore the influence of the power grid on shafting after the hydropower unit is connected to the grid.

## Figures and Tables

**Figure 1 sensors-22-04266-f001:**
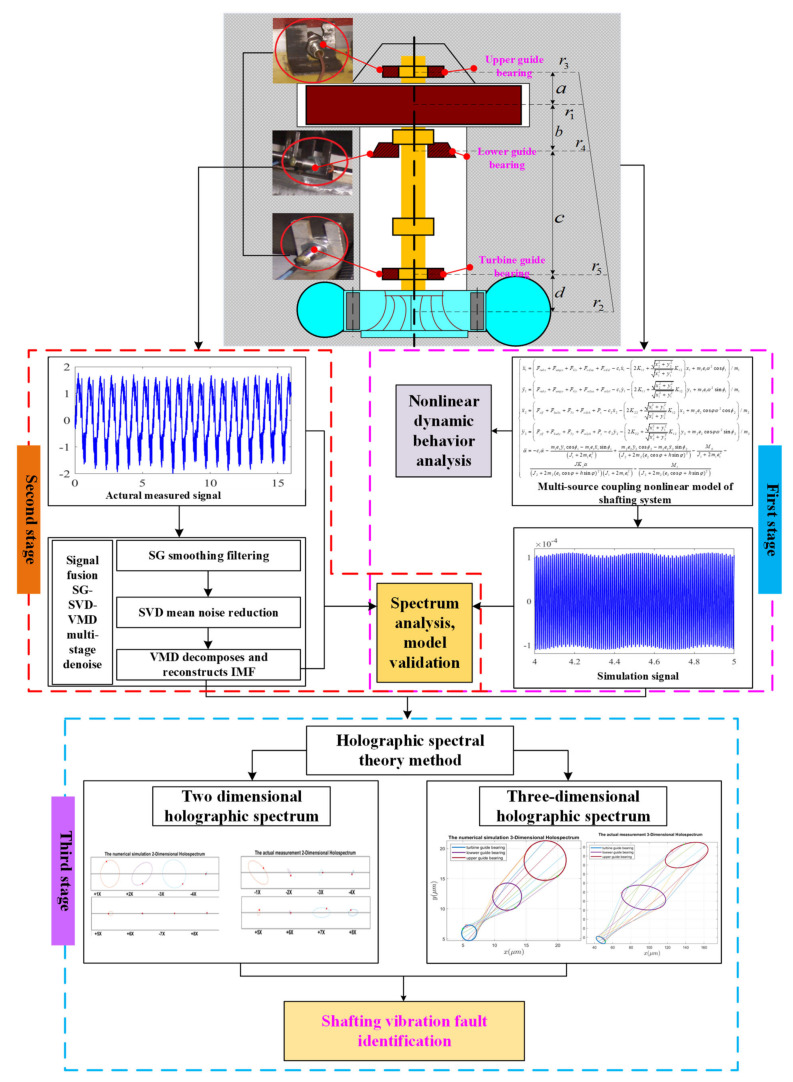
Shafting system vibration fault identification framework.

**Figure 2 sensors-22-04266-f002:**
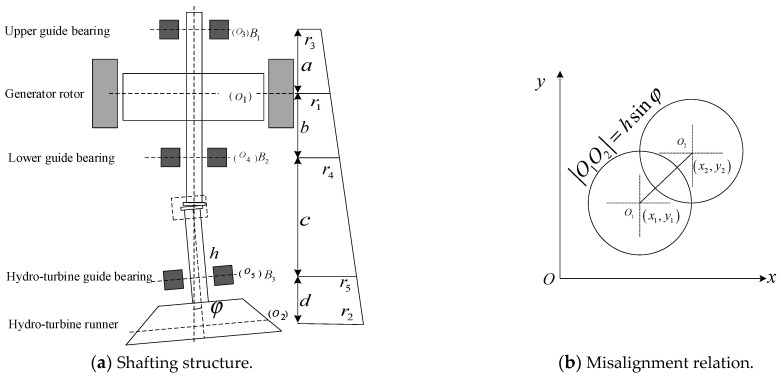
Misalignment shafting structure.

**Figure 3 sensors-22-04266-f003:**
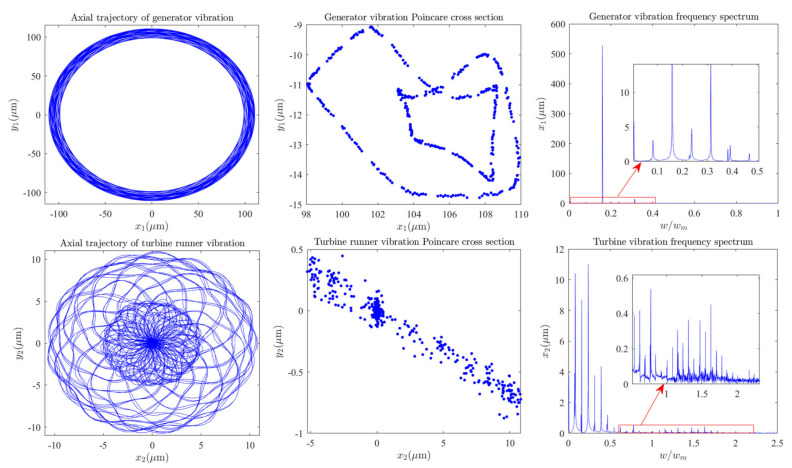
Dynamic characteristics of the shafting system.

**Figure 4 sensors-22-04266-f004:**
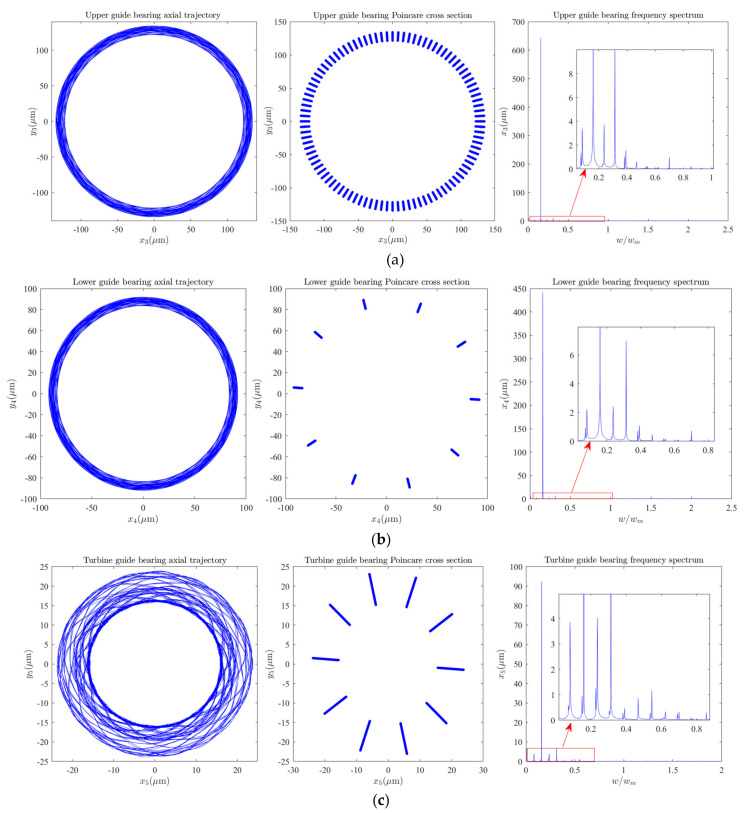
Dynamic characteristic of guide bearings: (**a**) Upper guide bearing dynamic characteristics, (**b**) Lower guide bearing dynamic characteristics, (**c**) Turbine guide bearing dynamic characteristics.

**Figure 5 sensors-22-04266-f005:**
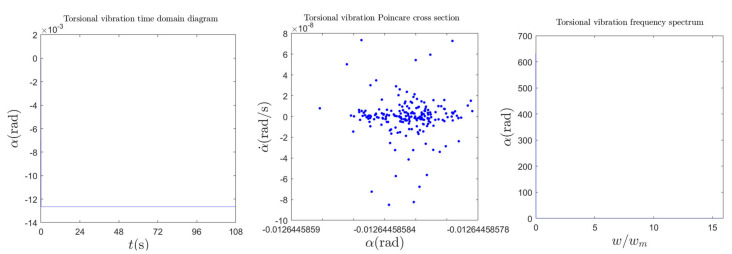
Dynamic behavior of torsional vibration of the shaft.

**Figure 6 sensors-22-04266-f006:**
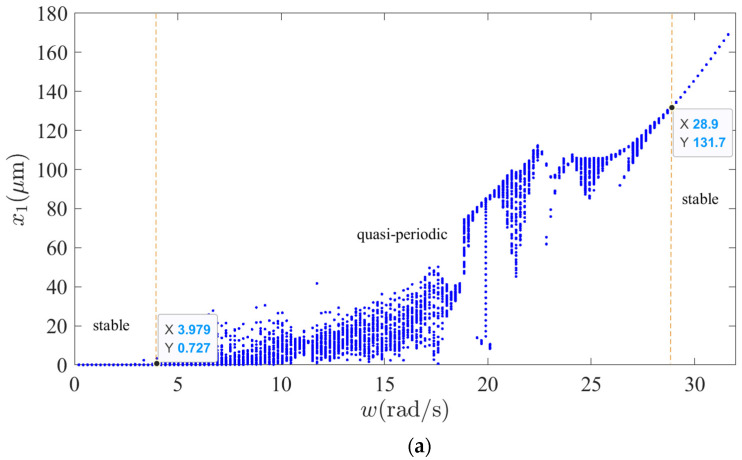
Shafting vibration nonlinear dynamic behavior: (**a**) Generator vibration bifurcation diagram, (**b**) Hydro-turbine vibration bifurcation diagram, (**c**) Upper guide bearing vibration bifurcation diagram, (**d**) Lower guide bearing vibration bifurcation diagram, (**e**) Hydro-turbine guide bearing vibration bifurcation diagram, (**f**) Torsional vibration bifurcation diagram.

**Figure 7 sensors-22-04266-f007:**
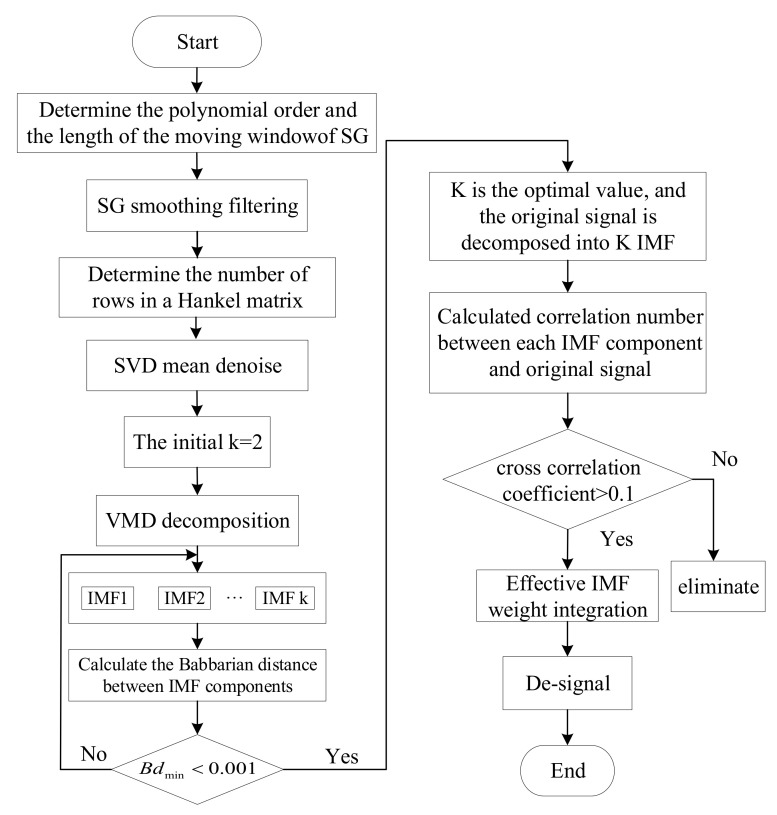
Signal multistage noise reduction process.

**Figure 8 sensors-22-04266-f008:**
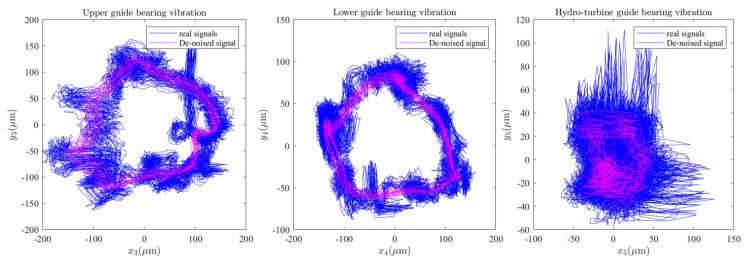
Shafting vibration before and after denoising.

**Figure 9 sensors-22-04266-f009:**
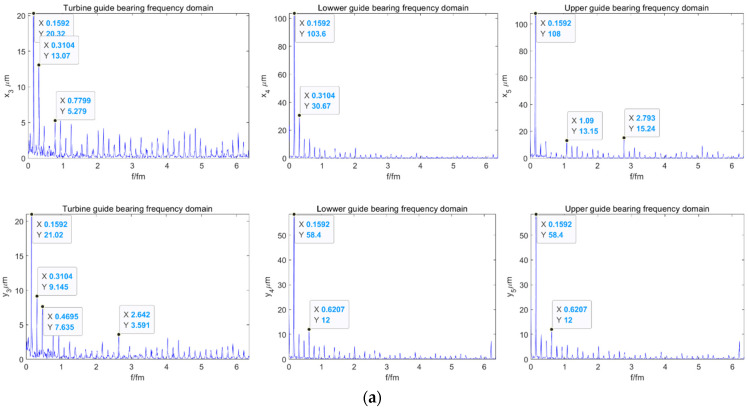
Spectrum diagram of the shafting vibration: (**a**) Spectrum diagram before shafting vibration denoising, (**b**) Spectrum diagram after shafting vibration denoising.

**Figure 10 sensors-22-04266-f010:**
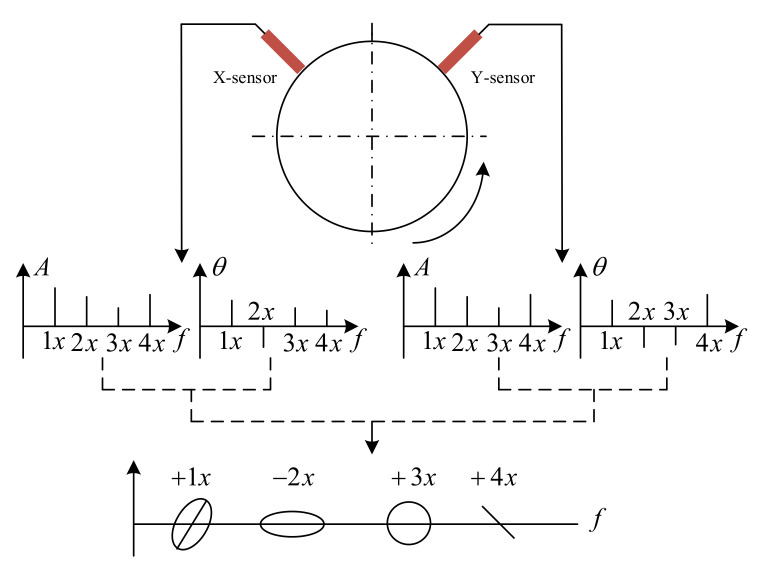
The 2D holospectrum schematic diagram.

**Figure 11 sensors-22-04266-f011:**
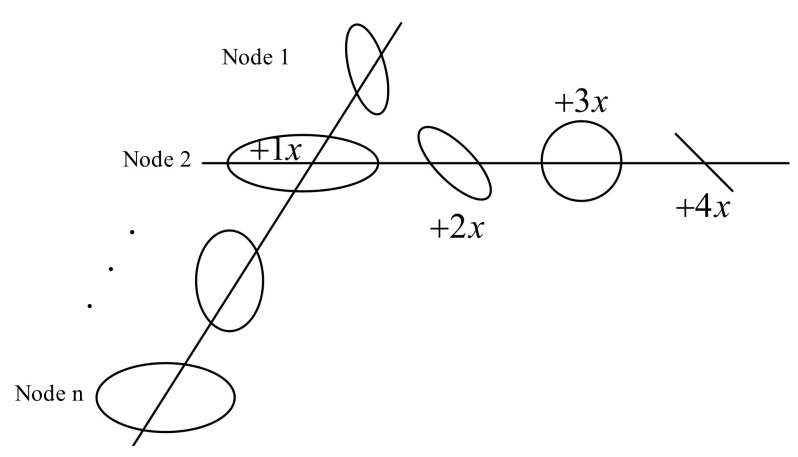
The 3D holospectrum.

**Figure 12 sensors-22-04266-f012:**
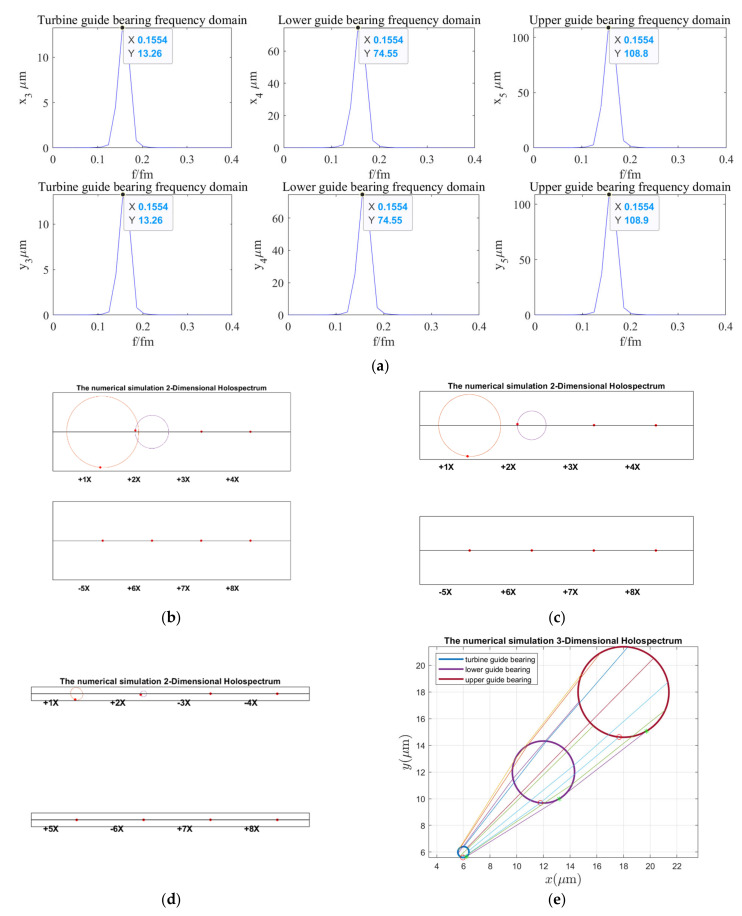
Holospectrum of shafting vibration of unbalance rotor: (**a**) Shafting vibration spectrogram, (**b**) Upper guide bearing 2D holospectrum, (**c**) Lower guide bearing 2D holospectrum, (**d**) Hydro-turbine guide bearing 2D holospectrum, (**e**) One-fold frequency 3D holospectrum.

**Figure 13 sensors-22-04266-f013:**
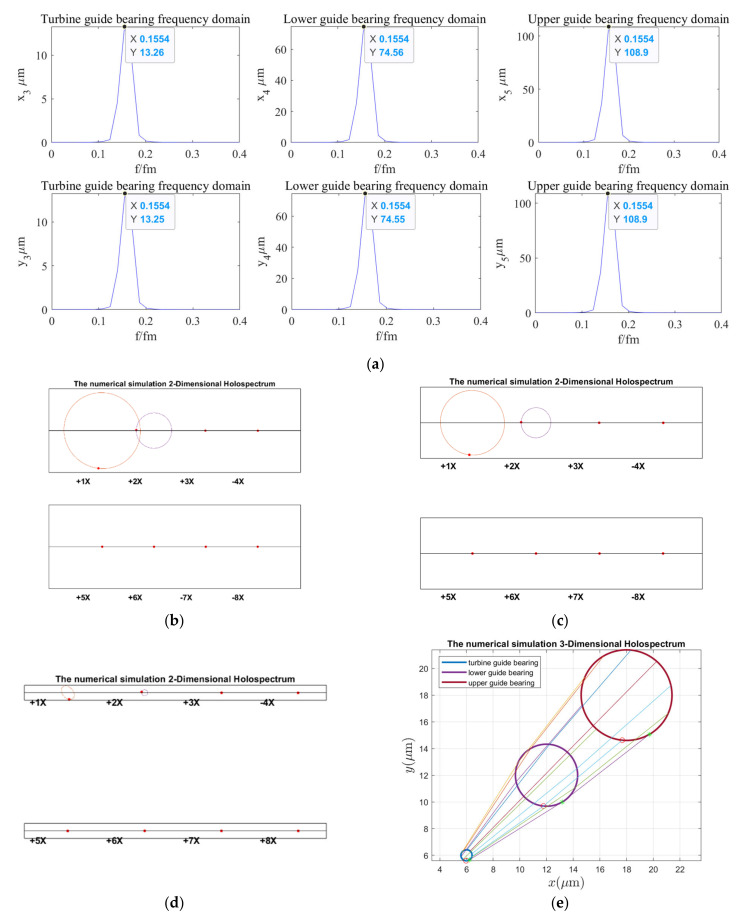
Holospectrum of shafting vibration of misaligned unbalance rotor: (**a**) Shafting vibration spectrogram, (**b**) Upper guide bearing 2D holospectrum, (**c**) Lower guide bearing 2D holospectrum, (**d**) Hydro-turbine guide bearing 2D holospectrum, (**e**) One-fold frequency 3D holospectrum.

**Figure 14 sensors-22-04266-f014:**
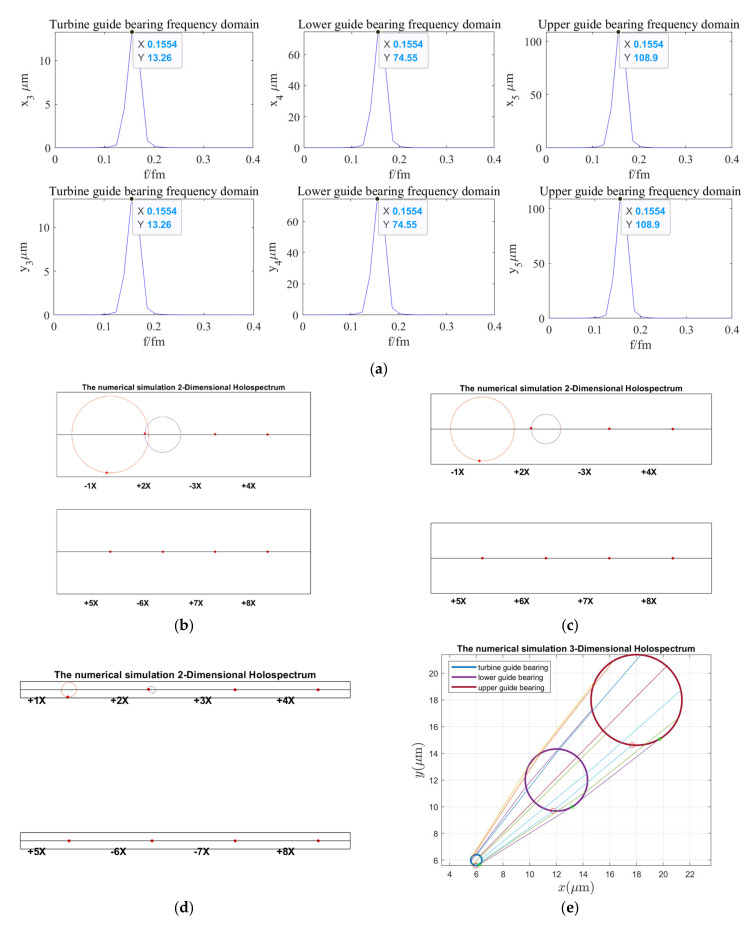
Holospectrum of shafting vibration of misaligned unbalance rotor, considering the oil film force: (**a**) Shafting vibration spectrogram, (**b**) Upper guide bearing 2D holospectrum, (**c**) Lower guide bearing 2D holospectrum, (**d**) Hydro-turbine guide bearing 2D holospectrum, (**e**) One-fold frequency 3D holospectrum.

**Figure 15 sensors-22-04266-f015:**
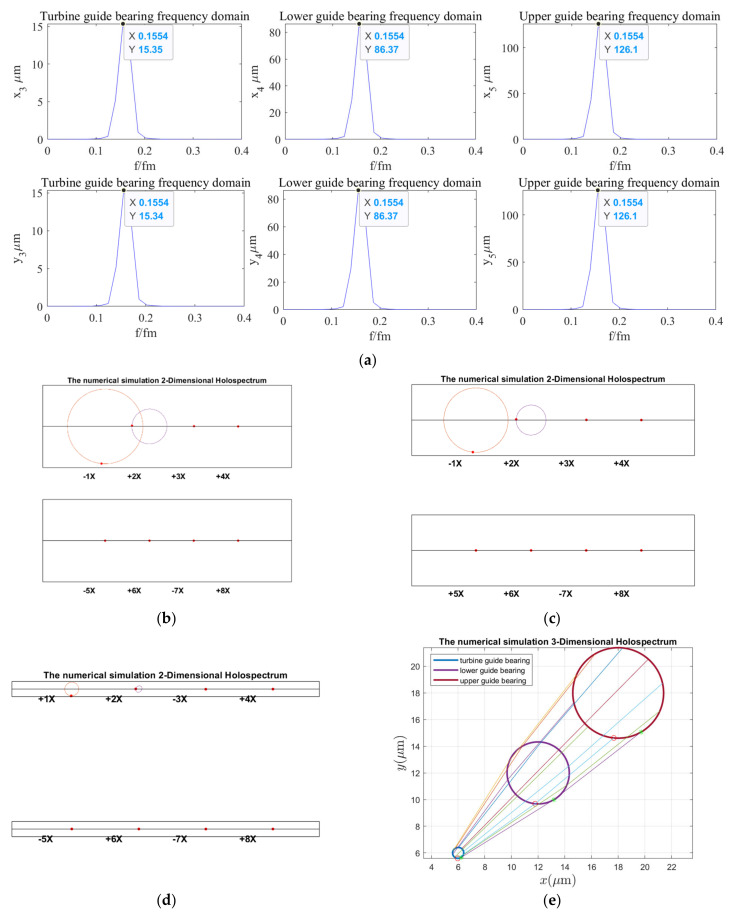
Holospectrum of shafting vibration of misaligned unbalance rotor consider oil film and RAW forces: (**a**) Shafting vibration spectrogram, (**b**) Upper guide bearing 2D holospectrum, (**c**) Lower guide bearing 2D holospectrum, (**d**) Hydro-turbine guide bearing 2D holospectrum, (**e**) One-fold frequency 3D holospectrum.

**Figure 16 sensors-22-04266-f016:**
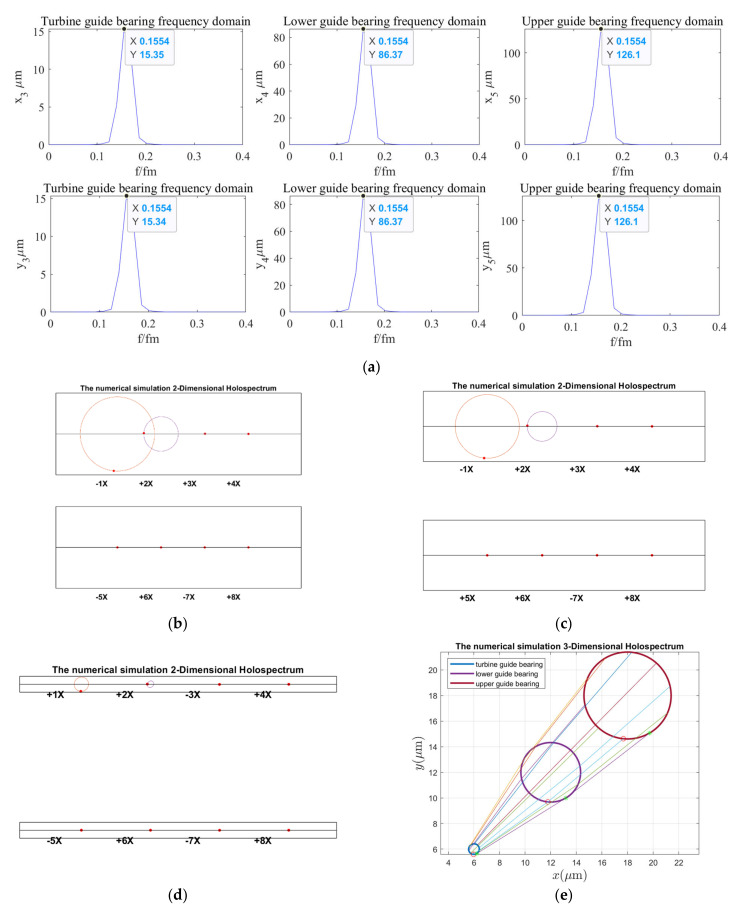
Holospectrum of shafting vibration of misaligned unbalance rotor, considering the oil film, RAW, and rub forces: (**a**) Shafting vibration spectrogram, (**b**) Upper guide bearing 2D holospectrum, (**c**) Lower guide bearing 2D holospectrum, (**d**) Hydro-turbine guide bearing 2D holospectrum, (**e**) One-fold frequency 3D holospectrum.

**Figure 17 sensors-22-04266-f017:**
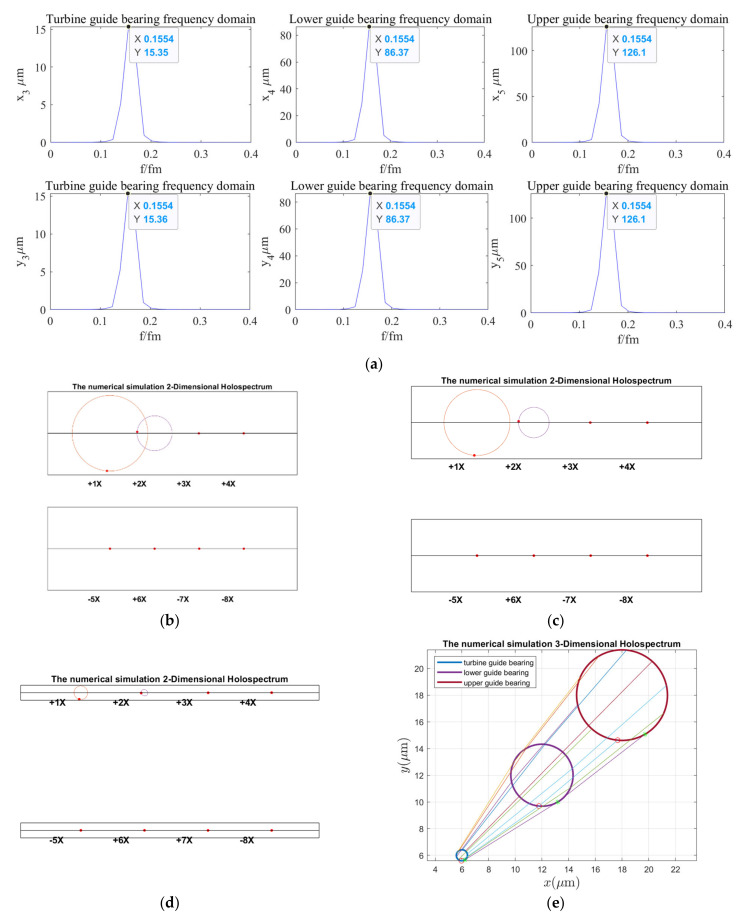
Holospectrum of shafting vibration of misaligned unbalance rotor, considering the oil film, RAW, rub, and UMP forces: (**a**) Shafting vibration spectrogram, (**b**) Upper guide bearing 2D holospectrum, (**c**) Lower guide bearing 2D holospectrum, (**d**) Hydro-turbine guide bearing 2D holospectrum, (**e**) One-fold frequency 3D holospectrum.

**Figure 18 sensors-22-04266-f018:**
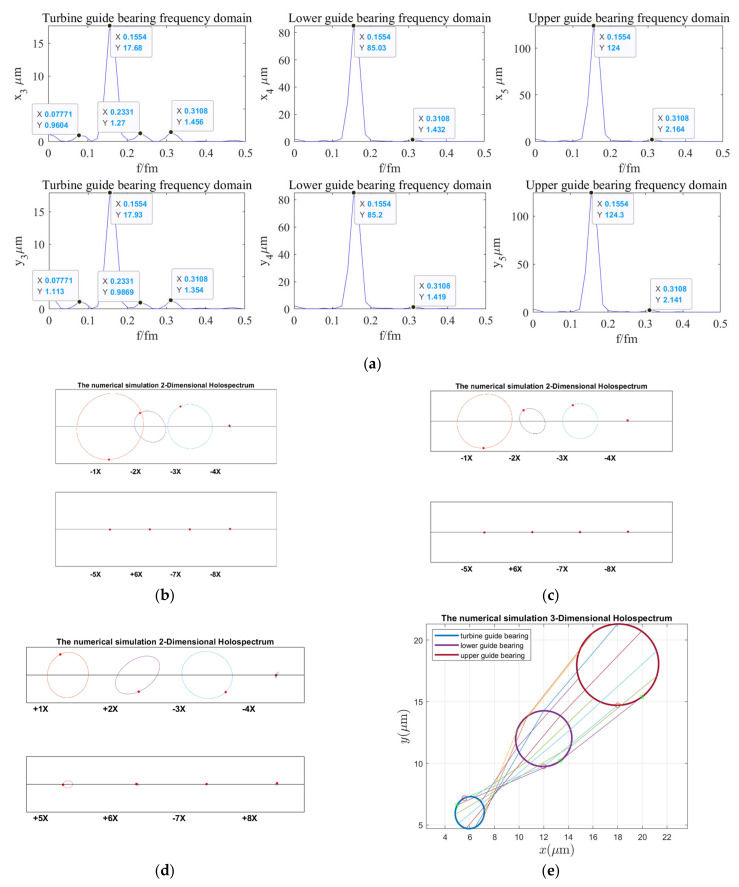
Holospectrum of shafting vibration of misaligned unbalance rotor, considering the oil film, RAW, rub, UMP, and turbine runner vortex eccentricity forces: (**a**) Shafting vibration spectrogram, (**b**) Upper guide bearing 2D holospectrum, (**c**) Lower guide bearing 2D holospectrum, (**d**) Hydro-turbine guide bearing 2D holospectrum, (**e**) One-fold frequency 3D holospectrum.

**Figure 19 sensors-22-04266-f019:**
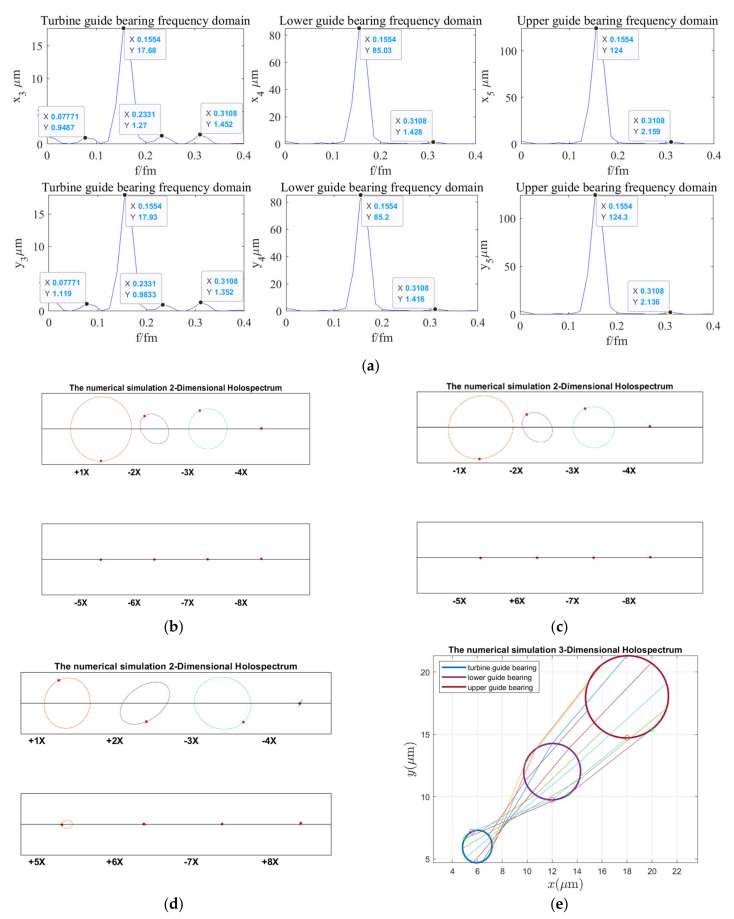
Holospectrum of shafting vibration of misaligned unbalance rotor, considering the oil film, RAW, rub, UMP, turbine runner vortex eccentricity, and hydraulic imbalance forces: (**a**) Shafting vibration spectrogram, (**b**) Upper guide bearing 2D holospectrum, (**c**) Lower guide bearing 2D holospectrum, (**d**) Hydro-turbine guide bearing 2D holospectrum, (**e**) One-fold frequency 3D holospectrum.

**Figure 20 sensors-22-04266-f020:**
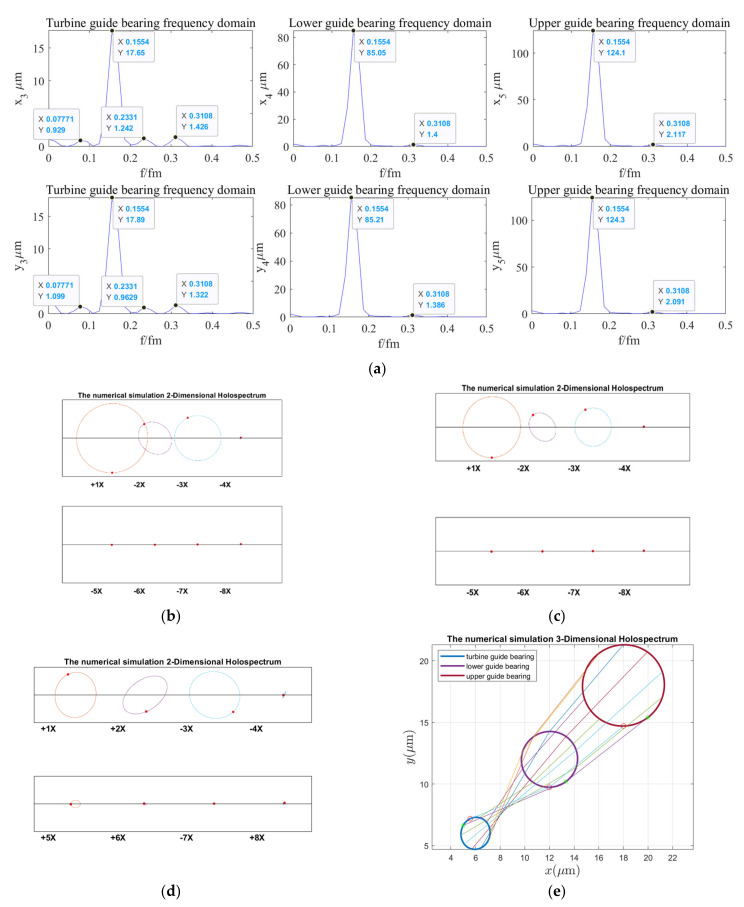
Shafting vibration holospectra of multi-vibration sources: (**a**) Shafting vibration spectrogram, (**b**) Upper guide bearing 2D holospectrum, (**c**) Lower guide bearing 2D holospectrum, (**d**) Hydro-turbine guide bearing 2D holospectrum, (**e**) One-fold frequency 3D holospectrum.

**Figure 21 sensors-22-04266-f021:**
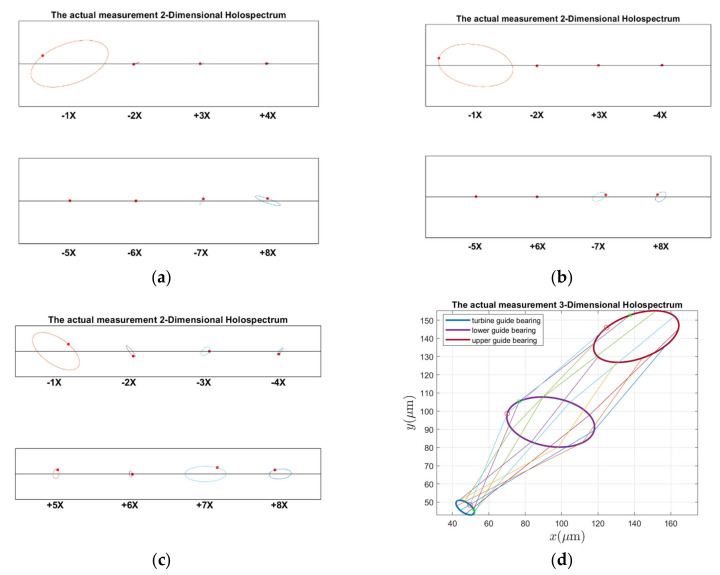
Shafting vibration holospectra of actual measurements: (**a**) Upper guide bearing 2D holospectrum, (**b**) Lower guide bearing 2D holospectrum, (**c**) Hydro-turbine guide bearing 2D holospectrum, (**d**) One-fold frequency 3D holospectrum.

**Figure 22 sensors-22-04266-f022:**
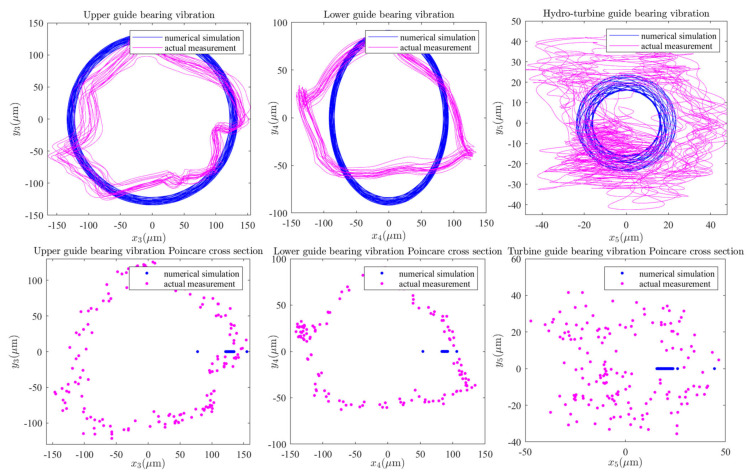
Guide bearing vibration axial center trajectories and Poincare cross-sections.

**Table 1 sensors-22-04266-t001:** Actual design parameters of a hydropower station.

Parameter	Value	Unit	Parameter	Value	Unit
$m_{1}$	565,432	kg	a	2.54	m
$m_{2}$	11,700	kg	b	2.04	m
$c_{1}$	$4.1\times 10^{6}$	N·s/m	c	8.19	m
$c_{2}$	$2.52\times 10^{6}$	N·s/m	d	1.765	m
$c_{t}$	1000	N·s/rad	$e_{1}$	0.0005	m
$k_{1}$	$1.85\times 10^{8}$	N/m	$e_{2}$	0.0005	m
$k_{2}$	$1.05\times 10^{8}$	N/m	$d_{H}$	1.15	m
$k_{3}$	$1.05\times 10^{10}$	N/m	$d_{B}$	0.3	m
$j_{1}$	$4.625\times 10^{6}$	kg·m^2^	$d_{1}$	7.44	m
$j_{2}$	$2\times 10^{5}$	kg·m^2^	L	2.927	m
E	$206\times 10^{9}$	Pa	D	7.44	m
Q	176.1	m^3^/s	$\delta_{0}$	0.0021	m
f	0.12		$\delta_{2}$	0.0035	m
$k_{r}$	6,100,000	N/m	l	10.3	m
g	9.81	m/s^−2^	h	6.045	m
$\mu_{2}$	0.11		$b_{d}$	0.994	m
$\mu_{0}$	0.6		$n_{r}$	250	r/min
Δp	$0.5\times 10^{6}$	Pa	$S_{gB}$	306	MW
$K_{z}$	0.8		$\beta_{1}$	16.1	°
$Ra_{0}$	2703	cm	$\beta_{2}$	14.2	°
$m_{0}$	−0.25		$\tau_{0}$	0.23	
$b_{0}$	3.5		$D_{1}$	5.34	m
$T_{0}$	0.0657		B	1	T
$r_{a}$	1.86	m	β	0.5	
$r_{2}$	2.67	m	$n_{0}$	0.066	

**Table 2 sensors-22-04266-t002:** The SNR and RMSE of each signal denoising method.

Methods	Index	*X* _3_	*Y* _3_	*X* _4_	*Y* _4_	*X* _5_	*Y* _5_
VMD	SNR	17.6353	16.0648	19.7347	16.3642	5.9473	4.8077
	RMSE	0.0818	0.0810	0.0570	0.0863	0.1828	0.2108
SG-VMD	SNR	16.0114	14.1497	15.2229	14.1068	5.5645	4.8988
	RMSE	0.0987	0.1010	0.0956	0.1118	0.1911	0.2086
SVD-VMD	SNR	17.5892	16.0299	16.8659	16.3438	8.5060	4.8702
	RMSE	0.0823	0.0813	0.0791	0.0864	0.1362	0.2093
SVD-SG-VMD	SNR	16.0286	13.7879	13.2089	14.0245	5.5583	4.9030
	RMSE	0.0985	0.1053	0.1205	0.1129	0.1912	0.2087
VMD-SG	SNR	17.6333	16.0305	19.7115	16.3774	5.8921	4.7716
	RMSE	0.0819	0.0813	0.0571	0.0861	0.1840	0.2117
VMD-SVD	SNR	17.5842	16.0022	19.6979	16.3266	5.8383	4.7849
	RMSE	0.0823	0.0816	0.0572	0.0866	0.1851	0. 2114
VMD-SVD-SG	SNR	17.6131	15.9934	19.7435	16.3497	5.8492	4.7694
	RMSE	0.0821	0.0817	0.0569	0.0864	0.1849	0.2118
VMD-SG-SVD	SNR	17.5872	15.9790	19.6888	16.3201	5.8118	4.7769
	RMSE	0.0823	0.0818	0.0572	0.0867	0.1857	0.2116
**SG-SVD-VMD**	**SNR**	**18.7629**	**16.7913**	**19.7437**	**16.3663**	**6.2178**	**4.9064**
	**RMSE**	**0.0719**	**0.0745**	**0.0568**	**0.0862**	**0.1772**	**0.2085**

## Data Availability

The data presented in this study are available on request from the corresponding author.

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
