# Peer review of "A Vibration Fault Identification Framework for Shafting Systems of Hydropower Units: Nonlinear Modeling, Signal Processing, and Holographic Identification"

_sensors, 2022, doi:10.3390/s22114266_

Round 1
Reviewer 1 Report
Dear authors,
Your title proposal
A vibration fault identification framework for shafting system 2 of hydropower units: nonlinear modeling, signal processing, 3 and holographic identification
It is scientifically interesting, but I still express the main elements that I have identified.
1) Regarding the methodology used by you in this article proposal, it is important to identify the boundary that separates theoretical and practical studies from a certain so-called field - State of the Art - and our contribution on the theoretical and practical side.
If we have a theoretical contribution, it is useful to present in detail the previous models and to emphasize our contribution through mathematical demonstrations, otherwise I do not see the point of loading an article with classical mathematical models.
If we have a case study that makes practical contributions, the narrative description of the equipment, the methods used is good to be compared with what we have so far in the field.
2) Previous research in the field is good to list, but the excessive addition of some elements from previous research I do not think is beneficial for a scientific contribution, because we have reproductible research instead of work in progress.
From page 19 in section 3. SG-SVD-VMD fusion signal denoising
We have a similarity with a previous publication.
From line 322 to line 410.
From an ethical point of view, I don't think it's fair.
https://iopscience.iop.org/article/10.1088/1742-6596/2108/1/012008
3) On the technical dating side, I recommend using the LaTeX / Tex template provided by MDPI.
I consider your proposal to be valuable, but it is good to review the issues found in point 2.
I am willing to review a modified version.
I have attached the file with the similarity.
I wish you success in your research!

Author Response
- Regarding the methodology used by you in this article proposal, it is important to identify the boundary that separates theoretical and practical studies from a certain so-called field - State of the Art - and our contribution on the theoretical and practical side. If we have a theoretical contribution, it is useful to present in detail the previous models and to emphasize our contribution through mathematical demonstrations, otherwise I do not see the point of loading an article with classical mathematical models. If we have a case study that makes practical contributions, the narrative description of the equipment, the methods used is good to be compared with what we have so far in the field.
Response: Thank you for your review of this submitted paper and valuable constructive advice. In this paper, the classical theories used in shafting modeling research is the Lagrange equation principle and holographic spectrum theory was used in fault identification, which was proposed by other scholars. The innovation of this paper is mainly reflected in the establishment of the vibration equation of shafting coupled with multiple vibration sources by using the Lagrange equation principle. At the same time, in order to verify the effectiveness of the established model, the signal denoising method is adopted to de-noise the measured signal and compare it with the simulation signal to verify the correctness of the model. Furthermore, the vibration fault of shafting is identified by holographic spectrum theory.
- Previous research in the field is good to list, but the excessive addition of some elements from previous research I do not think is beneficial for a scientific contribution, because we have reproductible research instead of work in progress. From page 19 in section 3. SG-SVD-VMD fusion signal denoising. We have a similarity with a previous publication.From line 322 to line 410.From an ethical point of view, I don't think it's fair. https://iopscience.iop.org/article/10.1088/1742-6596/2108/1/012008
Response: Thank you for your review of this submitted paper and valuable constructive advice. The SG-SVD-VMD multi-stage noise reduction method is proposed by the author himself and embodies good noise reduction performance. This article has changed it to a reference, modified the description in the correct place, and deleted some of the duplicate content. Details are as follows:
- SG-SVD-VMD fusion signal denoising
In this section, we use SG-SVD-VMD [34-35] multistage denoising method to de-noise actual measured signals of the upper, lower, and hydro-turbine bearings of a power station same as section 2.3. The specific process is shown in Fig. 7.
|
Figure 7. Signal multistage noise reduction process
The length of the moving window is 41. Meanwhile, set the number of rows for the reconstructed Hankel matrix to 8. The signal to noise ratio (SNR) and root mean square error(RMSE) of each method for shafting guide bearing actual measured signal were shown in Table 2. As can be seen from Table 2, after SG-SVD-VMD denoising, the shafting vibration data has higher SNR and smaller RMSE. The measured data of shafting vibration of a power station were de-noised at multiple levels, and the data before and after de-noising were shown in Fig. 8, and the spectrum diagram before and after shafting vibration denoising is shown in Fig. 9. Meanwhile, as can be seen from Fig.8 and Fig.9, the Shafting vibration condition monitoring data by SG-SVD-VMD multi-stage noise reduction, some noise components have been filtered out.
Table 2. The SNR and RMSE of each signal denoise method.
Methods |
Index |
X3 |
Y3 |
X4 |
Y4 |
X5 |
Y5 |
VMD |
SNR |
17.6353 |
16.0648 |
19.7347 |
16.3642 |
5.9473 |
4.8077 |
RMSE |
0.0818 |
0.0810 |
0.0570 |
0.0863 |
0.1828 |
0.2108 |
|
SG-VMD |
SNR |
16.0114 |
14.1497 |
15.2229 |
14.1068 |
5.5645 |
4.8988 |
RMSE |
0.0987 |
0.1010 |
0.0956 |
0.1118 |
0.1911 |
0.2086 |
|
SVD-VMD |
SNR |
17.5892 |
16.0299 |
16.8659 |
16.3438 |
8.5060 |
4.8702 |
RMSE |
0.0823 |
0.0813 |
0.0791 |
0.0864 |
0.1362 |
0.2093 |
|
SVD-SG-VMD |
SNR |
16.0286 |
13.7879 |
13.2089 |
14.0245 |
5.5583 |
4.9030 |
RMSE |
0.0985 |
0.1053 |
0.1205 |
0.1129 |
0.1912 |
0.2087 |
|
VMD-SG |
SNR |
17.6333 |
16.0305 |
19.7115 |
16.3774 |
5.8921 |
4.7716 |
RMSE |
0.0819 |
0.0813 |
0.0571 |
0.0861 |
0.1840 |
0.2117 |
|
VMD-SVD |
SNR |
17.5842 |
16.0022 |
19.6979 |
16.3266 |
5.8383 |
4.7849 |
RMSE |
0.0823 |
0.0816 |
0.0572 |
0.0866 |
0.1851 |
0. 2114 |
|
VMD-SVD-SG |
SNR |
17.6131 |
15.9934 |
19.7435 |
16.3497 |
5.8492 |
4.7694 |
RMSE |
0.0821 |
0.0817 |
0.0569 |
0.0864 |
0.1849 |
0.2118 |
|
VMD-SG-SVD |
SNR |
17.5872 |
15.9790 |
19.6888 |
16.3201 |
5.8118 |
4.7769 |
RMSE |
0.0823 |
0.0818 |
0.0572 |
0.0867 |
0.1857 |
0.2116 |
|
SG-SVD-VMD |
SNR |
18.7629 |
16.7913 |
19.7437 |
16.3663 |
6.2178 |
4.9064 |
RMSE |
0.0719 |
0.0745 |
0.0568 |
0.0862 |
0.1772 |
0.2085 |
|
Fig.8 Shafting vibration before and after de-noising |
|
(a) Spectrum diagram before shafting vibration denoising |
|
(b) Spectrum diagram after shafting vibration denoising |
Fig.9 Spectrum diagram of the shafting vibration |
[35] Shi Yousong, Zhou Jianzhong. Multistage noise reduction processing for vibration signal of hydropower units[J]. Journal of Physics: Conference Series,2021,2108(1).
- On the technical dating side, I recommend using the LaTeX / Tex template provided by MDPI. I consider your proposal to be valuable, but it is good to review the issues found in point 2. I am willing to review a modified version. I have attached the file with the similarity. I wish you success in your research!
Response: Thank you for your review of this submitted paper and valuable constructive advice. We have made a lot of modifications to the original manuscript, including modifying the correct description, modifying some wrong grammar, deleting some unnecessary content, and submitting the revised manuscript in the form of a template.

Reviewer 2 Report
This paper presents a shafting vibration fault identified framework, which includes three stages: nonlinear modeling, signal denoise, and holographic identification. Overall, the paper is well written and organized with a proper length. The contributions as well as the quality are both good. In addition, there are some points that are not very clear and should be addressed in the revised version:
1. Please update the references, especially for recent years. Meanwhile, analytical model-based fault identification framework is widely applied on electrical system fault diagnosis. For example the following references had given significant design results:
[1] Incipient winding fault detection and diagnosis for squirrel-cage induction motors equipped on CRH trains. ISA Transactions, 2020, 99: 488~495.
2. The innovation of this paper is not clear and it is difficult for readers to understand the main contributions of this paper. This part should be added in Introduction section.
3. The description of the existing work should be shorter in Introduction section. Furthermore, more descriptions of the proposed method are needed.
4. Labels in Fig.10 should be in English.
5. The reviewers recommend that more future work should be added on Conclusion Section.
Reviewer 3 Report
- This paper seems to be mainly based on modeling and simulation. Is there any specific experimental data to verify the proposed model and the proposed signal processing method? There is no relevant experimental description in the paper.
- The flowchart shown in Figure 1 is incomplete.
- The paper is too long, some well-known theories and methods can be simplified, such as signal processing methods SG, SVD, VMD and holographic spectrum, etc. It is recommended to shorten the content.
Reviewer 4 Report
Identification of shaft beats is a very important problem. The difficulty lies in the lack of sensors on many machines. Today, the use of such systems is mandatory according to ISO 19283 Condition monitoring and diagnostics of machines - Hydroelectric generating units. Sensor installation is specified in ISO 20816-5:2018 Mechanical vibration — Measurement and evaluation of machine vibration — Part 5: Machine sets in hydraulic power generating and pump-storage plants.
Paper drawbacks:
- There are green marks in the text, they should be removed.
- In Fig. 2, small formulas are not visible, the background in the upper part is bad
- Eq. 4 requires variables to be written, not all readers may know special notation for read correct
- Line 379-380 eq. and 439-440 need fix it
- Fig.10 upper coner fix inscription
- Eq.53 need explanation why "=1", yes, but very far down from the eq.
- Desirable comparison with experiment
- Strong recommended to issue according to the template, a lot
mistakes
Round 2
Reviewer 1 Report
Dear authors,
I have carefully read the changes you have made.
I further express my opinion on the current state of research and your contribution to it, as well as the use of materials from previous articles in the articles | subsequent publications.
Continuation of research and its dissemination does not require detailed repetition of previous studies.
I wish you success in your research.
Respectfully